# Learning on graphs using Orthonormal Representation is Statistically Consistent

**Rakesh S**
Department of Electrical Engineering
Indian Institute of Science
Bangalore, 560012, INDIA
rakeshsmysore@gmail.com

**Chiranjib Bhattacharyya**
Department of CSA
Indian Institute of Science
Bangalore, 560012, INDIA
chiru@csa.iisc.ernet.in

## Abstract

Existing research [4] suggests that embedding graphs on a unit sphere can be beneficial in learning labels on the vertices of a graph. However the choice of optimal embedding remains an open issue. *Orthonormal representation* of graphs, a class of embeddings over the unit sphere, was introduced by Lovász [2]. In this paper, we show that there exists orthonormal representations which are statistically consistent over a large class of graphs, including power law and random graphs. This result is achieved by extending the notion of consistency designed in the inductive setting to graph transduction. As part of the analysis, we explicitly derive relationships between the Rademacher complexity measure and structural properties of graphs, such as the chromatic number. We further show the fraction of vertices of a graph $G$, on $n$ nodes, that need to be labelled for the learning algorithm to be consistent, also known as labelled sample complexity, is $\Omega\left(\frac{\vartheta(G)}{n}\right)^{\frac{1}{4}}$ where $\vartheta(G)$ is the famous Lovász $\vartheta$ function of the graph. This, for the first time, relates labelled sample complexity to graph connectivity properties, such as the density of graphs. In the multiview setting, whenever individual views are expressed by a graph, it is a well known heuristic that a convex combination of Laplacians [7] tend to improve accuracy. The analysis presented here easily extends to Multiple graph transduction, and helps develop a sound statistical understanding of the heuristic, previously unavailable.

## 1 Introduction

In this paper we study the problem of graph transduction on a simple, undirected graph $G = (V, E)$, with vertex set $V = [n]$ and edge set $E \subseteq V \times V$. We consider individual vertices to be labelled with binary values, $\pm 1$. Without loss of generality we assume that the first $fn$ vertices are labelled, i.e., the set of labelled vertices is given by $S = [fn]$, where $f \in (0, 1)$. Let $\bar{S} = V \backslash S$ be the unlabelled vertex set, and let $\mathbf{y}_S$ and $\mathbf{y}_{\bar{S}}$ be the labels corresponding to subgraphs $S$ and $\bar{S}$ respectively.

Given $G$ and $\mathbf{y}_S$, the goal of graph transduction is to learn predictions $\hat{\mathbf{y}} \in \mathbb{R}^n$, such that $er_{\bar{S}}^{0\text{-}1}[\hat{\mathbf{y}}] = \sum_{j \in \bar{S}} \mathbb{1}[y_j \neq \bar{y}_j]$, $\bar{\mathbf{y}} = sgn(\hat{\mathbf{y}})$ is small. To aid further discussion we introduce some notations.

**Notation** Let $\mathcal{S}^{n-1} = \{\mathbf{u} \in \mathbb{R}^n | \|\mathbf{u}\|_2 = 1\}$ denote a $(n-1)$ dimensional sphere. Let $\mathbf{D}_n$, $\mathbf{S}_n$ and $\mathbf{S}_n^+$ denote a set of $n \times n$ diagonal, square symmetric and square symmetric positive semidefinite matrices respectively. Let $\mathbb{R}_+^n$ be a non-negative orthant. Let $\mathbf{1}_n \in \mathbb{R}^n$ denote a vector of all 1's. Let $[n] := \{1, \ldots, n\}$. For any $\mathbf{M} \in \mathbf{S}_n$, let $\lambda_1(\mathbf{M}) \geq \ldots \geq \lambda_n(\mathbf{M})$ denote the eigenvalues and $M_i$ denote the $i^{th}$ row of $\mathbf{M}$, $\forall i \in [n]$. We denote the adjacency matrix of a graph $G$ by $\mathbf{A}$. Let $d_i$ denote the degree of vertex $i \in [n]$, $d_i := A_i^\top \mathbf{1}_n$. Let $\mathbf{D} \in \mathbf{D}_n$, where

$D_{ii} = d_i, \forall i \in [n]$. We refer $\mathbf{I} - \mathbf{D}^{-\frac{1}{2}} \mathbf{A} \mathbf{D}^{-\frac{1}{2}}$ as the Laplacian, where $\mathbf{I}$ denotes the identity matrix. Let $\bar{G}$ denote the complement graph of $G$, with the adjacency matrix $\bar{\mathbf{A}} = \mathbf{1}_n \mathbf{1}_n^\top - \mathbf{I} - \mathbf{A}$. For $\mathbf{K} \in \mathbf{S}_n^+$ and $\mathbf{y} \in \{\pm 1\}^n$, the dual formulation of Support vector machine (SVM) is given by

$$\omega(\mathbf{K}, \mathbf{y}) = \max_{\alpha \in \mathbb{R}_+^n} g(\alpha, \mathbf{K}, \mathbf{y}) \left( = \sum_{i=1}^n \alpha_i - \frac{1}{2} \sum_{i,j=1}^n \alpha_i \alpha_j y_i y_j \mathbf{K}_{ij} \right).$$ Let $\mathcal{Y} = \bar{\mathcal{Y}} = \{\pm 1\}, \widehat{\mathcal{Y}} \subseteq \mathbb{R}$

be the label, prediction and soft-prediction spaces over $V$. Given a graph $G$ and labels $\mathbf{y} \in \mathcal{Y}^n$ on $V$, let $cut(\mathbf{A}, \mathbf{y}) := \sum_{y_i \neq y_j} A_{ij}$. We use $\ell : \mathcal{Y} \times \widehat{\mathcal{Y}} \to \mathbb{R}_+$ to denote any loss function. In particular, for $a \in \mathcal{Y}$, $b \in \widehat{\mathcal{Y}}$, let $\ell^{0\text{-}1}(a,b) = \mathbb{1}[ab < 0]$, $\ell^{hinge}(a,b) = (1-ab)_+$ [1] and $\ell^{ramp}(a,b) = min(1, (1-ab)_+)$ denote the 0-1, hinge and ramp loss respectively. The notations $O, o, \Omega, \Theta$ will denote standard measures defined in asymptotic analysis [14].

**Motivation** Regularization framework is a widely used tool for learning labels on the vertices of a graph [23, 4]

$$\min_{\hat{\mathbf{y}} \in \mathcal{Y}^n} \frac{1}{|S|} \sum_{i \in S} \ell(y_i, \hat{y}_i) + \lambda \hat{\mathbf{y}}^\top \mathbf{K}^{-1} \hat{\mathbf{y}} \tag{1}$$

where $\mathbf{K}$ is a kernel matrix and $\lambda > 0$ is an appropriately chosen regularization parameter. It was shown in [4] that the optimal $\hat{\mathbf{y}}^*$ satisfies the following generalization bound

$$\mathbb{E}_S \left[ er_{\bar{S}}^{0\text{-}1} [\hat{\mathbf{y}}^*] \right] \leq c_1 \inf_{\hat{\mathbf{y}} \in \mathcal{Y}^n} \left\{ er_V[\hat{\mathbf{y}}] + \lambda \hat{\mathbf{y}}^\top \mathbf{K}^{-1} \hat{\mathbf{y}} \right\} + c_2 \left( \frac{tr_p(\mathbf{K})}{\lambda |S|} \right)^p$$

where $er_H^{(\cdot)}[\hat{\mathbf{y}}] := \frac{1}{|H|} \sum_{i \in H} \ell^{(\cdot)}(y_i, \hat{y}_i)$, $H \subseteq V$ [2]; $tr_p(\mathbf{K}) = \left( \frac{1}{n} \sum_{i=1}^n \mathbf{K}_{ii}^p \right)^{1/p}$, $p > 0$ and $c_1, c_2$ are dependent on $\ell$. [4] argued that for good generalization, $tr_p(\mathbf{K})$ should be a constant, which motivated them to normalize the diagonal entries of $\mathbf{K}$. It is important to note that the set of normalized kernels is quite big and the above presented analysis gives little insight in choosing the optimal kernel from such a set. The important problem of consistency $\left( er_{\bar{S}} \to 0 \text{ as } n \to \infty \right.$, to be formally defined in Section 3$)$ of graph transduction algorithms was introduced in [5]. [5] showed that the formulation (1), when used with a laplacian dependent kernel, achieves a generalization error of $\mathbb{E}_S [er_{\bar{S}} [\hat{\mathbf{y}}^*]] = O\left( \sqrt{\frac{q}{nf}} \right)$, where $q$ is the number of pure components [3]. Though [5]'s algorithm is consistent for a small number of pure components, they achieve the above convergence rate by choosing $\lambda$ dependent on true labels of the unlabeled nodes, which is not practical [6].

In this paper, we formalize the notion of consistency of graph transduction algorithms and derive novel graph-dependent statistical estimates for the following formulation.

$$\Lambda_C(\mathbf{K}, \mathbf{y}_S) = \min_{\bar{\mathbf{y}}_j \in \bar{\mathcal{Y}}, j \in \bar{S}} \min_{\alpha \in \mathbb{R}_+^n} \frac{1}{2} \alpha^\top \mathbf{K} \alpha + C \sum_{i \in S} \ell(\hat{y}_i, y_i) + C \sum_{j \in \bar{S}} \ell(\hat{y}_j, \bar{y}_j) \tag{2}$$

where $\hat{y}_k = \sum_{i \in S} K_{ik} y_i \alpha_i + \sum_{j \in \bar{S}} K_{jk} \bar{y}_j \alpha_j, \ \forall k \in V$. If all the labels are observed then [22] showed that the above formulation is equivalent to (1). We note that the normalization step considered by [4] is equivalent to finding an embedding of a graph on a sphere. Thus, we study *orthonormal representations* of graphs [2], which define a rich class of graph embeddings on an unit sphere. We show that the formulation (2) working with orthonormal representations of graphs is statistically consistent over a large class of graphs, including random and power law graphs. In the sequel, we apply Rademacher complexity to orthonormal representations of graphs and derive novel graph-dependent transductive error bound. We also extend our analysis to study multiple graph transduction. More specifically, we make the following contributions.

**Contributions** The main contribution of this paper is that we show there exists orthonormal representations of graphs that are statistically consistent on a large class of graph families $\mathcal{G}_c$. For a special orthonormal representation—**LS** labelling, we show consistency on Erdös Rényi random graphs. Given a graph $G \in \mathcal{G}_c$, with a constant fraction of nodes labelled $f = O(1)$, we derive

an error convergence rate of $er_{\bar{S}}^{0\text{-}1} = O\left(\frac{\vartheta(G)}{n}\right)^{\frac{1}{4}}$, with high probability; where $\vartheta(G)$ is the Lovász $\vartheta$ function of the graph $G$. Existing work [5] showed an expected convergence rate of $O\left(\sqrt{\frac{q}{n}}\right)$, however $q$ is dependent on the true labels of the unlabelled nodes. Hence their bound cannot be computed explicitly [6]. We also apply Rademacher complexity measure to the function class associated with orthonormal representations and derive a tight bound relating to $\chi(G)$, the chromatic number of the graph $G$. We show that the Laplacian inverse [4] has $O(1)$ complexity on graphs with high connectivity, whereas **LS** labelling exhibits a complexity of $\Theta(n^{\frac{1}{4}})$. Experiments demonstrate superior performance of **LS** labelling on several real world datasets. We derive novel transductive error bound, relating to graph structural measures. Using our analysis, we show that observing labels of $\Omega\left(\frac{\vartheta(G)}{n}\right)^{\frac{1}{4}}$ fraction of the nodes is sufficient to achieve consistency. We also propose an efficient Multiple Kernel Learning (MKL) based algorithm, with generalization guarantees for multiple graph transduction. Experiments demonstrate improved performance in combining multiple graphs.

## 2 Preliminaries

**Orthonormal Representation:** [2] introduced the idea of orthonormal representations for the problem of embedding a graph on a unit sphere. More formally, an orthonormal representation of a simple, undirected graph $G = (V, E)$ with $V = [n]$, is a matrix $\mathbf{U} = [\mathbf{u}_1, \ldots, \mathbf{u}_n] \in \mathbb{R}^{d \times n}$ such that $\mathbf{u}_i^T \mathbf{u}_j = 0$ whenever $(i, j) \notin E$ and $\mathbf{u}_i \in \mathcal{S}^{d-1} \, \forall i \in [n]$.

Let $Lab(G)$ denote the set of all possible orthonormal representations of the graph $G$ given by $Lab(G) := \{\mathbf{U} | \mathbf{U} \text{ is an Orthonormal Representation}\}$. [1] recently introduced the notion of graph embedding to Machine Learning community and showed connections to graph kernel matrices. Consider the set of graph kernels $\mathcal{K}(G) := \{\mathbf{K} \in \mathbf{S}_n^+ | K_{ii} = 1, \forall i \in [n]; K_{ij} = 0, \forall (i, j) \notin E\}$. [1] showed that for every valid kernel $\mathbf{K} \in \mathcal{K}(G)$, there exists an orthonormal representation $\mathbf{U} \in Lab(G)$; and it is easy to see the other way, $\mathbf{K} = \mathbf{U}^\top \mathbf{U} \in \mathcal{K}(G)$. Thus, the two sets, $Lab(G)$ and $\mathcal{K}(G)$ are equivalent. Orthonormal representation is also associated with an interesting quantity, the Lovász number [2], defined as: $\vartheta(G) = 2\left(\min_{\mathbf{K} \in \mathcal{K}(G)} \omega(\mathbf{K}, \mathbf{1}_n)\right)$ [1]. $\vartheta$ function is a fundamental tool for combinatorial optimization and approximation algorithms for graphs.

**Lovász Sandwich Theorem:** [2] Given an undirected graph $G = (V, E)$, $I\left(\bar{G}\right) \leq \vartheta\left(\bar{G}\right) \leq \chi(G)$; where $I\left(\bar{G}\right)$ is the independent number of the complement graph $\bar{G}$.

## 3 Statistical Consistency of Graph Transduction Algorithms

In this section, we formalize the notion of consistency of graph transduction algorithms. Given a graph $G_n = (V_n, E_n)$ of $n$ nodes, with labels of subgraph $S_n \subseteq V_n$ observable, let $er_{\bar{S}_n}^* := \inf_{\tilde{\mathbf{y}} \in \bar{\mathcal{Y}}^n} er_{\bar{S}_n}[\tilde{\mathbf{y}}]$ denote the minimal unlabelled node set error. Consistency is a measure of the quality of the learning algorithm $\mathcal{A}$, comparing $er_{\bar{S}_n}[\hat{\mathbf{y}}]$ to $er_{\bar{S}_n}^*$, where $\hat{\mathbf{y}}$ are the predictions made by $\mathcal{A}$. A related notion of *loss consistency* has been extensively studied in literature [3, 12], which only show that the difference $er_{\bar{S}_n}[\hat{\mathbf{y}}] - er_{S_n}[\hat{\mathbf{y}}] \to 0$ as $n \to \infty$ [6]. This does not confirm the optimality of $\mathcal{A}$, that is $er_{\bar{S}_n}[\hat{\mathbf{y}}] \to er_{\bar{S}_n}^*$. Hence, a notion stronger than loss consistency is needed.

Let $G_n$ belong to a graph family $\mathcal{G}$, $\forall n$. Let $\Pi_f$ be the uniform distribution over the random draw of the labelled subgraph $S_n \subseteq V_n$, such that $|S_n| = fn$, $f \in (0, 1)$. As discussed earlier, we want the $\ell$-regret, $\mathcal{R}_{S_n}[\mathcal{A}] = er_{\bar{S}_n}[\hat{\mathbf{y}}] - er_{\bar{S}_n}^*$ to be small. Since, the labelled nodes are drawn randomly, there is a small probability that one gets an unrepresentative subgraph $S_n$. However, for large $n$, we want $\ell$-regret to be close to zero with high probability[4]. In other words, for every finite and fixed $n$, we want to have an estimate on the $\ell$-regret, which decreases as $n$ increases. We define the following notion of consistency of graph transduction algorithms to capture this requirement

**Definition 1.** Let $\mathcal{G}$ be a graph family and $f \in (0, 1)$ be fixed. Let $\mathcal{V} = \{(v_i, y_i, E_i)\}_{i=1}^{\infty}$ be an infinite sequence of labelled node set, where $y_i \in \mathcal{Y}$ and $E_i$ is the edge information of node $v_i$ with the previously observed nodes $v_1, \ldots, v_{i-1}$, $\forall i \geq 2$. Let $V_n$ be the first $n$ nodes in $\mathcal{V}$, and let

$G_n \in \mathcal{G}$ be the graph defined by $(V_n, E_1, \ldots, E_n)$. Let $S_n \subseteq V_n$, and let $\mathbf{y}_n$, $\mathbf{y}_{S_n}$ be the labels of $V_n$, $S_n$ respectively. A learning algorithm $\mathcal{A}$ when given $G_n$ and $\mathbf{y}_{S_n}$ returns soft-predictions $\hat{\mathbf{y}}$ is said to be $\ell$-consistent *w.r.t* $\mathcal{G}$ if, when the labelled subgraph $S_n$ are random drawn from $\Pi_f$, the $\ell$-regret converges in probability to zero, i.e., $\forall \epsilon > 0$

$$\mathbf{Pr}_{S_n \sim \Pi_f}\left[\mathcal{R}_{S_n}[\mathcal{A}] \geq \epsilon\right] \to 0 \qquad as \quad n \to \infty$$

In Section 6 we show that the kernel learning style algorithm (2) working with orthonormal representations is consistent on a large class of graph families. To the best of our knowledge, we are not aware of any literature which provide an explicit empirical error convergence rate and prove consistency of the graph transduction algorithm considered. Before we prove our main result, we gather useful tools—**a)** complexity measure, which reacts to the structural properties of the graph (Section 4); **b)** generalization analysis to bound $er_{\bar{S}}$ (Section 5). In the interest of space, we defer most of the proofs to the supplementary material[5].

# 4 Graph Complexity Measures

In this section we apply Rademacher complexity to orthonormal representations of graphs, and relate to the chromatic number. In particular, we study **LS** labelling, whose class complexity can be shown to be greater than that of the Laplacian inverse on a large class of graphs.

Let (2) be solved for $\mathbf{K} \in \mathcal{K}(G)$, and let $\mathbf{U} \in Lab(G)$ be the orthonormal representation corresponding to $\mathbf{K}$ (Section 2). Then by Representer's theorem, the classifier learnt by (2) is of the form $h = \mathbf{U}\beta$, $\beta \in \mathbb{R}^n$. We define Rademacher complexity of the function class associated with orthonormal representations

**Definition 2** (Rademacher Complexity). Given a graph $G = (V, E)$, with $V = [n]$; let $\mathbf{U} \in Lab(G)$ and $\bar{\mathcal{H}}_{\mathbf{U}} = \{h | h = \mathbf{U}\beta, \beta \in \mathbb{R}^n\}$ be the function class associated with $\mathbf{U}$. For $p \in (0, 1/2)$, let $\sigma = (\sigma_1, \ldots, \sigma_n)$ be a vector of *i.i.d.* random variables such that $\sigma_i \sim \{+1, -1, 0\}$ *w.p.* $p$, $p$ and $1 - 2p$ respectively. The Rademacher complexity of the graph $G$ defined by $\mathbf{U}$, $\bar{\mathcal{H}}_{\mathbf{U}}$ is given by

$R(\bar{\mathcal{H}}_{\mathbf{U}}, p) = \frac{1}{n}\mathbb{E}_{\sigma}\left[sup_{h \in \bar{\mathcal{H}}_{\mathbf{U}}} \sum_{i=1}^{n} \sigma_i \langle h, \mathbf{u}_i \rangle\right]$

The above definition is motivated from [9, 3]. This is an empirical complexity measure, suited for the transductive settings. We derive the following novel tight Rademacher bound

**Theorem 4.1.** Let $G = (V, E)$ be a simple, undirected graph with $V = [n]$, $\mathbf{U} \in Lab(G)$ and $p \in [1/n, 1/2]$. Let $\mathcal{H}_{\mathbf{U}} = \{h \mid h = \mathbf{U}\beta, \beta \in \mathbb{R}^n, \|\beta\|_2 \leq tC\sqrt{n}\}, C > 0$, $t \in [0, 1]$ and let $\mathbf{K} = \mathbf{U}^\top \mathbf{U} \in \mathcal{K}(G)$ be the graph-kernel corresponding to $\mathbf{U}$. The Rademacher complexity of graph $G$ defined by $\mathbf{U}$ is given by $R(\mathcal{H}_{\mathbf{U}}, p) = c_0 tC\sqrt{p\lambda_1(\mathbf{K})}$, where $1/2\sqrt{2} \leq c_0 \leq \sqrt{2}$ is a constant.

The above result provides a lower bound for the Rademacher complexity for any unit sphere graph embedding. While upper-bounds maybe available [9, 3] but, to the best of our knowledge, this is the first attempt at establishing lower bounds. The use of orthonormal representations allow us to relate class complexity measure to graph-structural properties.

**Corollary 4.2.** For $C, t, p = O(1)$, $R(\mathcal{H}_{\mathbf{U}}, p) = O(\sqrt{\chi(G)})$. (Suppl.)

Such connections between learning theory complexity measures and graph properties was previously unavailable [9, 3]. Corollary 4.2 suggests that there exists graph regularizers with class complexity as large as $O(\sqrt{\chi(G)})$, which motivate us to find substantially better regularizers. In particular, we investigate **LS** labelling [16]; given a graph $G$, **LS** labelling $\mathbf{K}_{LS} \in \mathcal{K}(G)$ is defined as

$$\mathbf{K}_{LS} = \frac{\mathbf{A}}{\rho} + \mathbf{I}, \ \rho \geq |\lambda_n(\mathbf{A})| \tag{3}$$

**LS** labellinghas high Rademacher complexity on a large class of graphs, in particular

**Corollary 4.3.** For a random graph $G(n, q), q \in [0, 1)$, where each edge is present independently *w.p.* $q$; for $C, t, q = O(1)$ the Rademacher complexity of the function class associated with **LS** labelling (3) is $\Theta(n^{\frac{1}{4}})$, with high probability. (Suppl.)

For the limiting case of complete graphs, we can show that Laplacian inverse [4], the most widely used graph regularizer has $O(1)$ complexity (Claim 2, Suppl.), thus indicating that it may be suboptimal for graphs with high connectivity. Experimental results illustrate our observation.

We derive a class complexity measure for unit sphere graph embeddings, which indicates the richness of the function class, and helps the learning algorithm to choose an effective embedding.

## 5 Generalization Error Bound

In the previous section, we applied Rademacher complexity to orthonormal representations. In this section we derive novel graph-dependent generalization error bounds, which will be used in Section 6. Following a similar proof technique as in [3], we propose the following error bound—

**Theorem 5.1.** Given a graph $G = (V, E)$, $V = [n]$ with $\mathbf{y} \in \mathcal{Y}^n$ being the unknown binary labels over $V$; let $\mathbf{U} \in Lab(G)$, and $\mathbf{K} \in \mathcal{K}(G)$ be the corresponding kernel. Let $\tilde{\mathcal{H}}_{\mathbf{U}} = \{h | h = \mathbf{U}\beta, \ \beta \in \mathbb{R}^n, \ \|\beta\|_\infty \leq C\}$, $C > 0$. Let $\ell$ be any loss function, bounded in $[0, B]$ and $L$-Lipschitz in its second argument. For $f \in (0, 1/2]^6$, let labels of subgraph $S \subseteq V$ be observable, $|S| = nf$. Let $\bar{S} = V \backslash S$. For any $\delta > 0$ and $h \in \tilde{\mathcal{H}}_{\mathbf{U}}$, with probability $\geq 1 - \delta$ over $S \sim \Pi_f$

$$er_{\bar{S}}[\hat{\mathbf{y}}] \leq er_S[\hat{\mathbf{y}}] + LC\sqrt{\frac{2\lambda_1(\mathbf{K})}{f(1-f)}} + \frac{c_1 B}{1-f}\sqrt{\frac{1}{nf}\log\frac{1}{\delta}} \qquad (4)$$

where $\hat{\mathbf{y}} = \mathbf{U}^\top h$ and $c_1 > 0$ is a constant. (Suppl.)

**Discussion**  Note that from [2], $\lambda_1(\mathbf{K}) \leq \chi(G)$ and $\chi(G)$ is in-turn bounded by the maximum degree of the graph [21]. Thus, if $L, B, f = O(1)$, then for sparse, degree bounded graphs; for the choice of parameter $C = \Theta(1/\sqrt{n})$, the slack term and the complexity term goes to zero as $n$ increases. Thus, making the bound useful. Examples include tree, cycle, path, star and $d$-regular (with $d = O(1)$). Such connections relating generalization error to graph properties was not available before. We exploit this novel connection to analyze graph transduction algorithms in Section 6. Also, in Section 7, we extend the above result to the problem of multiple graph transduction.

### 5.1 Max-margin Orthonormal Representation

To analyze $er_S^{0\text{-}1}$ relating to graph structural measure, the $\vartheta$ function, we study the maximum margin induced by any orthonormal representation, in an oracle setting.

We study a fully '*labelled graph*' $G = (V, E, \mathbf{y})$, where $\mathbf{y} \in \mathcal{Y}^n$ are the binary labels on the vertices $V$. Given any $\mathbf{U} \in Lab(G)$, the maximum margin classifier is computed by solving $\omega(\mathbf{K}, \mathbf{y}) = g(\alpha^*, \mathbf{K}, \mathbf{y})$ where $\mathbf{K} = \mathbf{U}^\top\mathbf{U} \in \mathcal{K}(G)$. It is interesting to note that knowing all the labels, the max-margin orthonormal representation can be computed by solving an SDP. More formally

**Definition 3.** Given a labelled graph $G = (V, E, \mathbf{y})$, where $V = [n]$ and $\mathbf{y} \in \mathcal{Y}^n$ are the binary labels on $V$, let $\bar{\mathcal{H}} = \bigcup_{\mathbf{U} \in Lab(G)} \tilde{\mathcal{H}}_{\mathbf{U}}$, where $\tilde{\mathcal{H}}_{\mathbf{U}} = \{h | h = \mathbf{U}\beta, \ \beta \in \mathbb{R}^n\}$. Let $\mathbf{K} \in \mathcal{K}(G)$ be the kernel corresponding to $\mathbf{U} \in Lab(G)$. The *max-margin orthonormal representation* associated with the kernel function is given by $\mathbf{K}_{mm} = \text{argmin}_{\mathbf{K} \in \mathcal{K}(G)} \omega(\mathbf{K}, \mathbf{y})$.

By definition, $\mathbf{K}_{mm}$ induces the largest margin amongst other orthonormal representations, and hence is optimal. The optimal margin has interesting connections to the Lovász $\vartheta$ function —

**Theorem 5.2.** Given a labelled graph $G = (V, E, \mathbf{y})$, with $V = [n]$ and $\mathbf{y} \in \mathcal{Y}^n$ being the binary labels on vertices. Let $\mathbf{K}_{mm}$ be as in Definition 3, then $\omega(\mathbf{K}_{mm}, \mathbf{y}) = \vartheta(G)/2$. (Suppl.)

Thus, knowing all the labels, computing $\mathbf{K}_{mm}$ is equivalent to solving the $\vartheta$ function. However, in the transductive setting, $\mathbf{K}_{mm}$ cannot be computed. Alternatively, we explore **LS** labelling (3), which gives a constant factor approximation to the optimal margin on a large class of graphs.

**Definition 4.** A class of labelled graphs $\mathcal{G} = \{G = (V, E, \mathbf{y})\}$ is said to be a *Labelled SVM-$\vartheta$ graph family*, if there exist a constant $\gamma > 1$ such that $\forall G \in \mathcal{G}, \omega(\mathbf{K}_{LS}, \mathbf{y}) \leq \gamma\omega(\mathbf{K}_{mm}, \mathbf{y})$.

**Algorithm 1**

---

**Input:** $\mathbf{U}$, $\mathbf{y}_S$ and $C > 0$.
Get $\alpha^*, \bar{\mathbf{y}}_{\bar{S}}^*$ by solving $\Lambda_C(\mathbf{K}, \mathbf{y}_S)$ (2) for $\ell^{hinge}$ and $\mathbf{K} = \mathbf{U}^\top \mathbf{U}$.
**Return:** $\hat{\mathbf{y}} = \mathbf{U}^\top h_S$, where $h_S = \mathbf{U}\mathbf{Y}\alpha^*$; $\mathbf{Y} \in D_n$, $\mathbf{Y} = y_i$, if $i \in S$, otherwise $\bar{y}_i^*$.

---

Such class of graphs are interesting, because one can get a constant factor approximation to the optimal margin, without the knowledge of the true labels e.g., Mixture of random graphs: $G = (V, E, \mathbf{y})$, with $\mathbf{y}^\top \mathbf{1}_n = 0$, $cut(\mathbf{A}, \mathbf{y}) \leq c\sqrt{n}$, for $c > 1$ being a constant and the subgraphs corresponding to the two classes form $G(n/2, 1/2)$ random graphs (Claim 3, Suppl.).

We relate the maximum geometric margin induced by orthonormal representations to the $\vartheta$ function of the graph. This allows us to derive novel graph dependent learning theory estimates.

## 6   Consistency of Orthonormal Representation of Graphs

Aggregating results from Section 4 and 5, we show that Algorithm 1 working with orthonormal representations of graphs is consistent on a large class of graph families. For every finite and fixed $n$, we derive an estimate on $er_{\bar{S}_n}^{0\text{-}1}$.

**Theorem 6.1.** For the setting as in Definition 1, let $f \in (0, 1/2]$ be fixed. Let $\hat{\mathbf{y}}$ be the predictions learnt by Algorithm 1 with inputs $\mathbf{U}_n \in Lab(G_n)$, $\mathbf{y}_{S_n}$ and $C^* = \left( \frac{\vartheta^2(G_n)(1-f)}{2^3 n^2 f \vartheta(\bar{G}_n)} \right)^{\frac{1}{4}}$. Then $\exists \mathbf{U}_n \in Lab(G_n)$, $\forall G_n$ such that with probability atleast $1 - \frac{1}{n}$ over $S_n \sim \Pi_f$

$$er_{\bar{S}_n}^{0\text{-}1}[\hat{\mathbf{y}}] = O\left( \left( \frac{\vartheta(G_n)}{f^3(1-f)n} \right)^{\frac{1}{4}} + \frac{1}{1-f} \sqrt{\frac{\log n}{nf}} \right)$$

*Proof.* Let $\mathbf{K}_n \in \mathcal{K}(G_n)$ be the max-margin kernel associated with $G_n$ (Definition 3), and let $\mathbf{U}_n \in Lab(G)$ be the corresponding orthonormal representation. Since $\ell^{ramp}$ is an upper bound on $\ell^{0\text{-}1}$, we concentrate on bounding $er_{\bar{S}_n}^{ramp}[\hat{\mathbf{y}}]$. Note that for any $C > 0$

$$C|S_n| \cdot er_{S_n}^{ramp}[\hat{\mathbf{y}}] \leq C|S_n| \cdot er_{S_n}^{hinge}[\hat{\mathbf{y}}] \leq \Lambda_C(\mathbf{K}_n, \mathbf{y}_{S_n})$$
$$\leq \Lambda_C(\mathbf{K}_n, \mathbf{y}_n) \leq \omega(\mathbf{K}_n, \mathbf{y}_n) = \frac{\vartheta(G_n)}{2}$$

The last inequality follows from Theorem 5.2. Note that for ramp loss $L = B = 1$; using Theorem 5.1 for $\delta = \frac{1}{n}$, it follows that with probability atleast $1 - \frac{1}{n}$ over the random draw of $S_n \sim \Pi_f$,

$$er_{\bar{S}_n}^{ramp}[\hat{\mathbf{y}}] \leq \frac{\vartheta(G_n)}{2Cnf} + C\sqrt{\frac{2\lambda_1(\mathbf{K}_n)}{f(1-f)}} + \frac{c_1}{1-f}\sqrt{\frac{\log n}{nf}} \qquad (5)$$

where $c_1 = O(1)$. Using $\lambda_1(\mathbf{K}_n) \leq \vartheta(\bar{G}_n)$ [2] and optimizing RHS for $C$, we get $C^* = \left( \frac{\vartheta^2(G_n)(1-f)}{2^3 n^2 f \vartheta(\bar{G}_n)} \right)^{\frac{1}{4}}$. Plugging back $C^*$ and using $\vartheta(G_n)\vartheta(\bar{G}_n) = n$ [2] proves the claim. $\qquad \square$

[5] showed that $\mathbb{E}_S\left[ er_{\bar{S}_n} \right] = O\left(\sqrt{\frac{q}{n}}\right)$. However, as noted in Section 1, the quantity $q$ is dependent on $\mathbf{y}_{\bar{S}_n}$, and hence their bounds cannot be computed explicitly [6].

We assume that the graph does not contain duplicate nodes with opposite labels, $er_{\bar{S}_n}^* = 0$. Thus, consistency follows from the fact that $\vartheta(G) \leq n$ and for large families of graphs it is $O(n^c)$ where $0 \leq c < 1$. This theorem points to the fact that if $f = O(1)$, then by Definition 1, Algorithm 1 is $\ell^{0\text{-}1}$- consistency over such class of graph families. Examples include

**Power-law graphs:** Graphs where the degree sequence follows a power law distribution. We show that $\vartheta(\bar{G}) = O(\sqrt{n})$ for naturally occurring power law graphs (Claim 4, Suppl.). Thus, working with the complement graph $(\bar{G})$, makes Algorithm 1 consistent.

**Random graphs:** For $G(n,q)$ graphs, $q = O(1)$; with high probability $\vartheta(G(n,q)) = \Theta(\sqrt{n})$ [13].

Note that choosing $\mathbf{K}_n$ for various graph families is difficult. Alternatively, for Labelled SVM-$\vartheta$ graph family (Definition 4), if Lovász $\vartheta$ function is sub-linear, then for the choice of **LS** labelling, Algorithm 1 is $\ell^{0\text{-}1}$ consistent. Examples include Mixture of random graphs (Section 5.1). Furthermore, we analyze the fraction of labelled nodes to be observed, such that Algorithm 1 is consistent.

**Corollary 6.2** (Labelled Sample Complexity). Given a graph family $\mathcal{G}_c$, such that $\vartheta(G_n) = O(n^c)$, $\forall G_n \in \mathcal{G}_c$ where $0 \le c < 1$. For $C = C^*$ as in Theorem 6.1; $\frac{1}{2}\left(\frac{\vartheta(G_n)}{n}\right)^{1/3-\varepsilon}$, $\varepsilon > 0$ fraction of labelled nodes is sufficient for Algorithm 1 to be $\ell^{0\text{-}1}$-consistent *w.r.t.* $\mathcal{G}_c$.

The proof directly follows from Theorem 6.1. As a consequence of the above result, we can argue that for sparse graphs ($\vartheta(G)$ is large) one would need a larger fraction of nodes labelled, but for denser graphs ($\vartheta(G)$ is small) even a smaller fraction of nodes being labelled suffices. Such connections relating sample complexity and graph properties was not available before.

To end this section, we discuss on the possible extensions to the inductive setting (Claim 5, Suppl.)— we can show that that the uniform convergence of $er_{\bar{S}}$ to $er_S$ in the transductive setting (for $f = 1/2$) is a necessary and sufficient condition for the uniform convergence of $er_S$ to the generalization error. Thus, the results presented here can be extended to the supervised setting. Furthermore, combining Theorem 5.1 with the results of [9], we can also extend our results to the semi-supervised setting.

# 7 Multiple Graph Transduction

Many real world problems can be posed as learning on multiple graphs [19, **?**]. Existing algorithms for single graph transduction [10, 15] cannot be trivially extended to the new setting. It is a well known heuristic that taking a convex combination of Laplacian improves classification performance [7], however the underlying principle is not well understood. We propose an efficient MKL style algorithm with generalization guarantees. Formally, the problem of multiple graph transduction is—

**Problem 1.** Given $\mathbb{G} = \{G^{(1)}, \ldots, G^{(m)}\}$ a set of simple, undirected graphs $G^{(k)} = (V, E^{(k)})$, defined on a common vertex set $V = [n]$. Without loss of generality we assume that the first $fn$ vertices are labelled, i.e., the set of labelled vertices is given by $S = [fn]$, where $f \in (0, 1)$. Let $\bar{S} = V \backslash S$ be the unlabelled node set. Let $\mathbf{y}_S, \mathbf{y}_{\bar{S}}$ be the labels of $S, \bar{S}$ respectively. Given $\mathbb{G}$ and labels $\mathbf{y}_S$, the goal is to accurately predict the labels of $\mathbf{y}_{\bar{S}}$.

Let $\mathbb{K} = \{\mathbf{K}^{(1)}, \ldots, \mathbf{K}^{(m)}\}$ be the set of kernels corresponding to graphs $\mathbb{G}$; $\mathbf{K}^{(k)} \in \mathcal{K}(G^{(k)}), \forall k \in [m]$. We propose the following MKL style formulation for multiple graph transduction

$$\Psi_C(\mathbb{K}, \mathbf{y}_S) = \min_{\eta \in \mathbb{R}_+^m, \|\eta\|_1 = 1} \; \min_{\bar{y}_j \in \bar{\mathcal{Y}}, \forall j \in \bar{S}} \; \max_{\alpha \in \mathbb{R}_+^n, \|\alpha\|_\infty \le C} \; g\left(\alpha, \sum_{k=1}^m \eta_k \mathbf{K}^{(k)}, [\mathbf{y}_S, \bar{\mathbf{y}}_{\bar{S}}]\right) \quad (6)$$

Extending our analysis from Section 5, we propose the following error bound

**Theorem 7.1.** For the setting as in Problem 1, let $f \in (0, 1/2]^7$ and $\mathbb{K} = \{\mathbf{K}^{(1)}, \ldots, \mathbf{K}^{(m)}\}$, $\mathbf{K}^{(k)} \in \mathcal{K}(G^{(k)})$, $\forall k \in [m]$. Let $\alpha^*, \eta^*, \bar{\mathbf{y}}_{\bar{S}}^*$ be the solution to $\Psi_C(\mathbb{K}, \mathbf{y}_S)$ (6). Let $\hat{\mathbf{y}} = \sum_{i=1}^m \eta_k^* \mathbf{K}^{(k)} \bar{\mathbf{Y}} \alpha^*$, where $\bar{\mathbf{Y}} \in D_n$, $\bar{\mathbf{Y}}_{ii} = y_i$ if $i \in S$, otherwise $\bar{y}_i^*$. Then, for any $\delta > 0$, with probability $\ge 1 - \delta$ over the choice of $S \subseteq V$ such that $|S| = nf$

$$er_{\bar{S}}^{0\text{-}1}[\hat{\mathbf{y}}] \le \frac{\bar{\Psi}(\mathbb{K}, \mathbf{y})}{Cnf} + C\sqrt{\frac{2\vartheta(\bar{G}^\cup)}{f(1-f)}} + \frac{c_1}{1-f}\sqrt{\frac{1}{nf}\log\frac{1}{\delta}}$$

where $c_1 = O(1)$, $\bar{\Psi}(\mathbb{K}, \mathbf{y}) = \min_{k \in [m]} \omega(\mathbf{K}^{(k)}, \mathbf{y})$ and $G^\cup$ is the union of graphs $\mathbb{G}^8$. (Suppl.)

The above result gives us the ability for the first time to analyze generalization performance of multiple graph transduction algorithms. The expression $\bar{\Psi}(\mathbb{K}, \mathbf{y})$ suggests that combining multiple graphs should improve performance over considering individual graphs separately. Similar to Section 6,

we can show that if *one* of the graph families $\mathcal{G}^{(l)}, l \in [m]$ of $\mathbb{G}$ obey $\vartheta(G_n^{(l)}) = O(n^c)$, $0 \le c < 1$; $G_n^{(l)} \in \mathcal{G}^{(l)}$, then there exists orthonormal representations $\mathbb{K}$, such that the MKL style algorithm optimizing for (6) is $\ell^{0\text{-}1}$-consistent over $\mathbb{G}$ (Claim 6, Suppl.). We can also show that combining graphs improves labelled sample complexity (Claim 7, Suppl.). This is a first attempt in developing a statistical understanding for the problem of multiple graph transduction.

## 8 Experimental results

We conduct two sets of experiments[9].

Table 1: Superior performance of **LS** labelling.

| Dataset | LS-lab | Un-Lap | N-Lap | KS-Lap |
|---|---|---|---|---|
| AuralSonar* | **76.5** | 68.1 | 66.7 | 69.2 |
| Yeast-SW-5-7* | **60.4** | 54.1 | 52.9 | 53.3 |
| Yeast-SW-5-12* | **78.6** | 61.2 | 60.5 | 64.3 |
| Yeast-SW-7-12* | **76.5** | 64.0 | 59.5 | 63.1 |
| Diabetes† | **73.1** | 68.3 | 68.6 | 68.5 |
| Fourclass† | **73.3** | 69.3 | 71.2 | 71.8 |

**Superior performance of LS labelling:** We use two datasets—similarity matrices* from [11] and RBF kernel[10] as similarity matrices for the UCI datasets†[8]. We built an unweighted graph by thresholding the similarity matrices about the mean. Let $\mathbf{L} = \mathbf{D} - \mathbf{A}$. For the regularized formulation (1), with 10% of labelled nodes observable, we test four types of kernel matrices—**LS** labelling(LS-lab), $(\lambda_1 \mathbf{I} + \mathbf{L})^{-1}$ (Un-Lap), $(\lambda_2 \mathbf{I} + \mathbf{D}^{-1/2}\mathbf{L}\mathbf{D}^{-1/2})^{-1}$ (N-Lap) and **K**-Scaling (KS-Lap) [4]. We choose the parameters $\lambda, \lambda_1$ and $\lambda_2$ by cross validation. Table 1 summarizes the results. Each entry is accuracy in % *w.r.t.* 0-1 loss, and the results were averaged over 100 iterations. Since we are thresholding by mean, the graphs have high connectivity. Thus, from Corollary 4.3, the function class associated with **LS** labellingis rich and expressive, and hence it outperforms previously proposed regularizers.

**Graph transduction across Multiple-views:** Learning on mutli-view data has been of recent interest [18]. Following a similar line of attack, we pose the problem of classification on multi-view data as multiple graph transduction. We investigate the recently launched Google dataset [17], which contains multiple views of video game YouTube videos, consisting of 13 feature types of auditory (Aud), visual (Vis) and textual (Txt) description. Each video is labelled one of 30 classes. For each of the views we construct similarity matrices using cosine distance and threshold about the mean to obtain

Table 2: Multiple Graphs Transduction. Each entry is accuracy in %.

| Graph | 1vs2 | 1vs3 | 1vs4 | 2vs3 | 2vs4 | 3vs4 |
|---|---|---|---|---|---|---|
| Aud | 62.8 | 64.8 | 68.3 | 59.3 | 50.8 | 61.5 |
| Vis | 68.9 | 65.6 | 68.9 | 69.1 | 70.3 | 75.1 |
| Txt | 68.7 | 59.2 | 64.8 | 64.6 | 60.9 | 65.4 |
| Unn | 69.7 | 60.3 | 52.7 | 62.7 | 67.4 | 62.5 |
| Maj | 72.7 | 75.2 | 80.5 | 65.4 | 62.6 | 77.4 |
| Int | 80.6 | 83.6 | 86.0 | 90.9 | 75.3 | 91.8 |
| MV | **98.9** | **93.4** | **95.6** | **97.7** | **87.7** | **98.8** |

unweighted graphs. We considered 20% of the data to be labelled. We show results on pair-wise classification for the first four classes. As a natural way of combining graphs, we compared our algorithm (6) (MV) with union (Unn), intersection (Int) and majority (Maj)[11] of graphs. We used **LS** labelling as the graph-kernel and (2) was used to solve single graph transduction. Table 2 summarizes the results, averaged over 20 iterations. We also state top accuracy in each of the views for comparison. As expected from our analysis in Theorem 7.1, we observe that combining multiple graphs significantly improves classification accuracy.

## 9 Conclusion

For the problem of graph transduction, we show that there exists orthonormal representations that are consistent over a large class of graphs. We also note that the Laplacian inverse regularizer is suboptimal on graphs with high connectivity, and alternatively show that **LS** labellingis not only consistent, but also exhibits high Rademacher complexity on a large class of graphs. Using our analysis, we also develop a sound statistical understanding of the improved classification performance in combining multiple graphs.

## Footnotes

[1] $(a)_+ = \max(a, 0)$.

[2] We drop the argument $\hat{\mathbf{y}}$, when implicit from the context.

[3] Pure component is a connected subgraph, where all the nodes in the subgraph have the same label.

[4] If $\mathcal{G}$ is not deterministic (e.g., Erdös Réyni), then there is small probability that one gets an unrepresentative graph, in which case we want the $\ell$-regret to be close to zero with high probability over $G_n \sim \mathcal{G}$.

[5]mllab.csa.iisc.ernet.in/rakeshs/nips14/suppl.pdf

[6]We can generalize our result for $f \in (0, 1)$, but for the simplicity of the proof we assume $f \in (0, 1/2]$. This is also true in practice, where the number of labelled examples is usually very small.

[7]As in Theorem 5.1, we can generalize our results for $f \in (0, 1)$.

[8]$G^\cup = (V, E^\cup)$, where $(i, j) \in E^\cup$ if edge $(i, j)$ is present in atleast one of the graphs $G^{(k)} \in \mathbb{G}, k \in [m]$.

[9]Relevant resources at: `mllab.csa.iisc.ernet.in/rakeshs/nips14`

[10]The $(i, j)^{th}$ entry of an RBF kernel is given by $exp\left(\frac{-\|\mathbf{x}_i - \mathbf{x}_j\|^2}{2\sigma^2}\right)$. We set $\sigma$ to the mean distance.

[11]Majority graph is a graph where an edge $(i, j)$ is present, if a majority of the graphs have the edge $(i, j)$.

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
