[Supplementary Material]

# Learning on graphs using Orthonormal Representation is Statistically Consistent

**Rakesh S**
Department of Electrical Engineering
Indian Institute of Science
Bangalore, 560012, INDIA
rakeshsmysore@gmail.com

**Chiranjib Bhattacharyya**
Department of CSA
Indian Institute of Science
Bangalore, 560012, INDIA
chiru@csa.iisc.ernet.in

## 1   Preliminaries

For completeness, we state/prove some of the non-trivial results used for proving results of the paper.

### 1.1   Asymptotic Notations [7]

For non-negative functions $f_1(n)$ and $f_2(n)$

- $f_1(n) = O(f_2(n)) \implies \exists n_0$ and a constant $c > 0$ such that $\forall n > n_0$, $f_1(n) \leq c f_2(n)$.
- $f_1(n) = \Omega(f_2(n)) \implies \exists n_0$ and a constant $c > 0$ such that $\forall n > n_0$, $f_1(n) \geq c f_2(n)$.
- $f_1(n) = \Theta(f_2(n))$ iff $f_1(n) = O(f_2(n))$ and $f_1(n) = \Omega(f_2(n))$.

### 1.2   Properties of $\vartheta$ for a labelled graph

For a labelled graph $G = (V, E, \mathbf{y})$, where nodes have binary labels +1/-1, we prove some interesting properties of the Lovász $\vartheta$ function, which will be used to prove examples of Labelled SVM-$\vartheta$ graphs (Definition 4). Let $G_+$ and $G_-$ be graphs corresponding to the two classes, defined as

$$G_+ = (V_+, E_+), \ V_+ = \{i \in V | y_i = +1\} \text{ and } (i,j) \in E_+ \text{ iff } (i,j) \in E \ \& \ y_i = y_j = +1 \quad (1)$$

Similarly, we can define $G_-$ as follows

$$G_- = (V_-, E_-), \ V_- = \{i \in V | y_i = -1\} \text{ and } (i,j) \in E_- \text{ iff } (i,j) \in E \ \& \ y_i = y_j = -1 \quad (2)$$

Let $G' = (V, E')$ be a pure graph, where $(i,j) \in E'$ iff $(i,j) \in E$ and $y_i = y_j$. We note an alternate definition of Lovász $\vartheta$ function

**Definition 1** ([11]).  For a simple, undirected graph G=(V,E),

$$\vartheta(G) = \min_{\mathbf{c} \in \mathbf{S}^{d-1}, \ \mathbf{U} \in Lab(G)} \max_{i \in V} \frac{1}{\left(\mathbf{c}^\top \mathbf{u}_i\right)^2} \quad (3)$$

We state the following bounds on the Lovász $\vartheta$ function

**Lemma 1.1.**  $\vartheta(G) \leq \vartheta(G') = \vartheta(G_+) + \vartheta(G_-)$

*Proof.*  The first inequality follows from the Definition 1 and the fact that $Lab(G') \subseteq Lab(G)$.

Without loss of generality, we assume that the nodes of the graph are ordered such that the nodes having positive labels appear first. Let $n_1 = \{i \in V | y_i = +1\}$ and $n_2 = n - n_1$. Note that

$\mathbf{K} \in \mathcal{K}(G')$ takes the form $[\mathbf{K}_1, \mathbf{0}_{n_1 \times n_2}; \ \mathbf{0}_{n_2 \times n_1}, \mathbf{K}_2]$, where $\mathbf{K}_1 \in \mathcal{K}(G_+)$ and $\mathbf{K}_2 \in \mathcal{K}(G_-)$. For any $\mathbf{K} \in \mathcal{K}(G')$, note that

$$\omega(\mathbf{K}, \mathbf{1}_n) = \max_{\alpha \in \mathbb{R}_+^n} \sum_{i=1}^n \alpha_i - \frac{1}{2} \sum_{i,j=1}^n \alpha_i \alpha_j K_{ij}$$

$$= \max_{\alpha \in \mathbb{R}_+^n} \sum_{i \in V_+} \alpha_i + \sum_{j \in V_-} \alpha_j - \frac{1}{2} \sum_{i,j \in V_+} \alpha_i \alpha_j K_{ij} - \frac{1}{2} \sum_{i,j \in V_-} \alpha_i \alpha_j K_{ij}$$

$$= \omega(\mathbf{K}_1, \mathbf{1}_{n_1}) + \omega(\mathbf{K}_2, \mathbf{1}_{n_2})$$

Thus, the result follows from the definition of $\vartheta$ function Section (2).

$$\frac{1}{2} \vartheta(G') = \min_{\mathbf{K} \in \mathcal{K}(G')} \omega(\mathbf{K}, \mathbf{1}_n) = \min_{\mathbf{K}_1 \in \mathcal{K}(G_+)} \omega(\mathbf{K}_1, \mathbf{1}_{n_1}) + \min_{\mathbf{K}_2 \in \mathcal{K}(G_-)} \omega(\mathbf{K}_2, \mathbf{1}_{n_2})$$

$$= \frac{1}{2} \Big( \vartheta(G_+) + \vartheta(G_-) \Big)$$

$\square$

**Lemma 1.2.** $\vartheta(G) \geq \max \{ \vartheta(G_+), \vartheta(G_-) \}$.

*Proof.* Proof follows from Definition 1.

$$\vartheta(G_+) = \min_{\mathbf{U} \in Lab(G_+)} \min_{\mathbf{c} \in \mathcal{S}^{d-1}} \max_{i \in V_+} \frac{1}{\left( \mathbf{c}^\top \mathbf{u}_i \right)^2}$$

$$\leq \min_{\mathbf{U} \in Lab(G)} \min_{\mathbf{c} \in \mathcal{S}^{d-1}} \max_{i \in V_+} \frac{1}{\left( \mathbf{c}^\top \mathbf{u}_i \right)^2}$$

$$\leq \min_{\mathbf{U} \in Lab(G)} \min_{\mathbf{c} \in \mathcal{S}^{d-1}} \max_{i \in V} \frac{1}{\left( \mathbf{c}^\top \mathbf{u}_i \right)^2} = \vartheta(G)$$

Similarly, can prove $\vartheta(G) \geq \vartheta(G_-)$. $\square$

Note that the above results can also be extended to any partition of the vertex set $V$ into $V_1, \dots, V_k$, $\cup_{i=1}^k = V$, $V_i \cap V_j = \phi$, $i \neq j$.

## 1.3 Martingale and Concentration inequalities

To bound the difference between a random variable and its expectation, Doob's martingale sequence [13] is a standard tool used in statistics –

**Definition 2** (Martingale). Let $X_1^n := (X_1, \dots, X_n)$ be a sequence of random variables. The sequence $T_0^n := (T_0, \dots, T_n)$ is called a *martingale w.r.t.* the underlying sequence $X_1^n$, if for any $i \in [n]$, $T_i$ is a function of $X_1^i$ and $\mathbb{E}_{X_i}[T_i | X_1^{i-1}] = T_{i-1}$.

For the martingale process above, we state the following concentration inequality.

**Lemma 1.3** ([13]). Let $T_0^n$ be a martingale *w.r.t.* $X_1^n$. Let $x_1^n = (x_1, \dots, x_n)$ be the vector of possible values of the random variables $X_1, \dots, X_n$. Let

$$r_i(x_1^{i-1}) := \sup_{x_i} \left[ T_i : X_1^{i-1} = x_1^{i-1}, X_i = x_i \right] - \inf_{x_i} \left[ T_i : X_1^{i-1} = x_1^{i-1}, X_i = x_i \right]$$

Let $\tilde{r} := \sup_{x_1^n} \sum_{i=1}^n \left( r_i(x_1^{i-1}) \right)^2$. Then, $\mathbf{Pr}_{X_1^n} \left[ T_n - T_0 \geq \epsilon \right] \leq \exp \left( -\frac{2\epsilon^2}{\tilde{r}} \right)$.

We note the following alternative definition of Rademacher variables, borrowed from [8]

**Definition 3** (Pairwise Rademacher variables). Let $\tau = \{ \tau_i = (\tau_{i,1}, \tau_{i,2}) \}_{i=1}^n$ be *i.i.d* random variables defined as:

$$\tau_i = (\tau_{i,1}, \tau_{i,2}) = \begin{cases} \left( \frac{1}{1-f}, \frac{1}{f} \right), & \text{w.p. } f(1-f); \\ \left( -\frac{1}{f}, -\frac{1}{1-f} \right), & \text{w.p. } f(1-f); \\ \left( -\frac{1}{f}, \frac{1}{f} \right), & \text{w.p. } f^2; \\ \left( \frac{1}{1-f}, -\frac{1}{1-f} \right), & \text{w.p. } (1-f)^2. \end{cases}$$

The above tools will be used to prove graph dependent generalization error bound Theorem 5.1.

We use the tools developed in this section to prove all the results of the paper in Section 2.

# 2 Proof of results in paper

## Section 4

We prove the following technical lemma to lower bound the expectation of square root of a non-negative random variable, using first and second order moments.

**Claim 1.** For any non-negative random variable $\mathbf{X}$

$$\mathbb{E}\left[\sqrt{\mathbf{X}}\right] \geq \sqrt{\mathbb{E}[\mathbf{X}]}\left[1 - \frac{Var[\mathbf{X}]}{2\mathbb{E}^2[\mathbf{X}]}\right]$$

*Proof.* For any $x, a \geq 0$, we prove

$$\sqrt{x} \geq \sqrt{a} + \frac{x-a}{2\sqrt{a}} - \frac{(x-a)^2}{2a\sqrt{a}} \tag{4}$$

Note that the result follows by choosing $x = \mathbf{X}$, $a = \mathbb{E}[\mathbf{X}]$ and taking expectation over (4). By simplification of (4), we get $2a\sqrt{ax} \geq 3ax - x^2$. Writing $b = \sqrt{x}$ and dividing by $b$ gives $2a\sqrt{a} \geq 3ab - b^3$. Note that for $a \geq 0$, function $g(b) = 3ab - b^3 - 2a\sqrt{a}$, $b \geq 0$ is concave with maximum is attained at $b = \sqrt{a}$ and $g(\sqrt{a}) = 0$. Thus $g(b) \leq 0, \forall b \geq 0$. $\qquad\square$

**Proof of Theorem 4.1.** For any $\sigma$, $\sup_{h \in \mathcal{H}_{\mathbf{U}}} \sum_{i=1}^{n} \sigma_i \langle h, \mathbf{u}_i \rangle = \sup_{h \in \mathcal{H}_{\mathbf{U}}} \left\langle h, \sum_{i=1}^{n} \sigma_i \mathbf{u}_i \right\rangle = tC\sqrt{n\lambda_1(\mathbf{K})}\left\|\sum_{i=1}^{n} \sigma_i \mathbf{u}_i\right\|$. The last equality from optimality over supremum and the norm constraint - $\max_{h \in \mathcal{H}_{\mathbf{U}}} \|h\| = \max_{\beta \in \mathbb{R}^n, \|\beta\|_2 \leq tC\sqrt{n}} \sqrt{\beta^\top \mathbf{K}\beta} = tC\sqrt{n\lambda_1(\mathbf{K})}$. Now, taking expectation over $\sigma$, one obtains

$$R(\mathcal{H}_{\mathbf{U}}, p) = tC\sqrt{\frac{\lambda_1(\mathbf{K})}{n}}\mathbb{E}_\sigma\left[\left\|\sum_{i=1}^{n} \sigma_i \mathbf{u}_i\right\|\right] = tC\sqrt{\frac{\lambda_1(\mathbf{K})}{n}}\mathbb{E}_\sigma\left[\sqrt{\sigma^\top \mathbf{K}\sigma}\right] \tag{5}$$

Using Jensen's inequality, $\mathbb{E}_\sigma[\sqrt{\sigma^\top \mathbf{K}\sigma}]$ can now be upper bounded by $\sqrt{\mathbb{E}_\sigma[\sigma^\top \mathbf{K}\sigma]}$. Further, using the independence of $\sigma$, the expectation evaluates to $2p\sum_{i=1}^{n} K_{ii} = 2pn$. Thus, $R(\mathcal{H}_{\mathbf{U}}, p) \leq tC\sqrt{2p\lambda_1(\mathbf{K})}$. For any non-negative random variable $\mathbf{X}$, note that $\mathbb{E}\left[\sqrt{\mathbf{X}}\right] \geq \sqrt{\mathbb{E}[\mathbf{X}]}\left[1 - \frac{Var[\mathbf{X}]}{2\mathbb{E}^2[\mathbf{X}]}\right]$ (Claim 1, Suppl.) Thus, for the random variable $\sigma^\top \mathbf{K}\sigma$ we have

$$\mathbb{E}\left[\sqrt{\sigma^\top \mathbf{K}\sigma}\right] \geq \sqrt{2np}\left[1 - \frac{2np + (2p)^2(n(n-1) + \sum_{i \neq j} K_{ij}^2 - n^2)}{8n^2p^2}\right] \geq \sqrt{2np}\left[1 - \frac{3}{4}\right]$$

The last inequality follows from the fact that $\sum_{i \neq j} K_{ij}^2 \leq \sum_{i=1}^{n} \lambda_i^2(\mathbf{K}) \leq \left[\sum_{i=1}^{n} \lambda_i(\mathbf{K})\right]^2 \leq n^2$ and $p \in [1/n, 1/2]$. Plugging in (5) proves $R(\mathcal{H}_{\mathbf{U}}, p) \geq \frac{tC}{2\sqrt{2}}\sqrt{p\lambda_1(\mathbf{K})}$, and hence the result. $\qquad\square$

**Proof of Corollary 4.2.** Using Theorem 4.1, $R(\mathcal{H}_{\mathbf{U}}, p) = O(\sqrt{\lambda_1(\mathbf{K})})$. From the definition of $\vartheta(\bar{G})$ as in [11], it follows that $\lambda_1(\mathbf{K}) \leq \vartheta(\bar{G})$. Finally, using Sandwich Theorem (Section 2) proves the claim. $\qquad\square$

We prove the following bound on the spectral norm of **LS** labelling of a random graph

**Proof of Corollary 4.3.** For $G(n, q)$ graphs, [9] showed that with probability $1 - e^{-n^c}$, $c > 0$, $\lambda_1(A) = nq(1 + o(1))$ and $|\lambda_n(A)| \leq 2\sqrt{nq(1-q)}$. For $q = O(1)$, choosing $\rho = \sqrt{n}$ makes $\mathbf{K}_{LS}$ a positive semi-definite matrix, and clearly $\lambda_1(\mathbf{K}_{LS}) = \Theta(\sqrt{n})$. Thus, proving the result. $\qquad\square$

We prove that the function class associated with Laplacian inverse is restrictive

**Claim 2.** For a complete graph $K_n$ of size $n$, where every pair of nodes is connected by an edge; Laplacian inverse (6) has class complexity of $O(1)$.

*Proof.* Laplacian inverse of a graph is given by [3]

$$\mathbf{K}_{Lap}^\dagger = \sum_{i:\lambda_i(\mathbf{K}_{Lap})>0} \frac{1}{\lambda_i(\mathbf{K}_{Lap})} \mathbf{w}_i \mathbf{w}_i^\top \tag{6}$$

where $\mathbf{K}_{Lap} = \mathbf{I} - \mathbf{D}^{-1/2}\mathbf{A}\mathbf{D}^{-1/2} = \sum_{i=1}^n \lambda_i(\mathbf{K}_{Lap})\mathbf{w}_i\mathbf{w}_i^\top$. Let $\mathbf{V}_{Lap} = [\mathbf{v}_1, \ldots, \mathbf{v}_n]$ be the feature mapping corresponding to Laplacian inverse kernel $\mathbf{K}_{Lap}^\dagger$; $(\mathbf{K}_{Lap}^\dagger)_{ij} = \langle \mathbf{v}_i, \mathbf{v}_j \rangle, \forall i, j \in [n]$. Let $\mathcal{H}_{Lap}^\dagger = \{h | h = \sum_{i=1}^n \beta_i \mathbf{v}_i,\ \beta \in \mathbb{R}^n,\ \|\beta\|_2 \leq tC\sqrt{n}\},\ C > 0,\ t \in [0,1]$. We follow a similar proof technique as in Theorem 4.1. For any $\sigma$,

$$\sup_{h \in \mathcal{H}_{Lap}^\dagger} \sum_{i=1}^n \sigma_i \langle h, \mathbf{v}_i \rangle = \sup_{h \in \mathcal{H}_{Lap}^\dagger} \left\langle h, \sum_{i=1}^n \sigma_i \mathbf{v}_i \right\rangle = tC\sqrt{n\lambda_1(\mathbf{K})} \left\| \sum_{i=1}^n \sigma_i \mathbf{v}_i \right\|$$

The last equality from optimality over supremum and the norm constraint - $\max_{h \in \mathcal{H}_{Lap}^\dagger} \|h\| = \max_{\beta \in \mathbb{R}^n, \|\beta\|_2 \leq tC\sqrt{n}} \sqrt{\beta^\top \mathbf{K}_{Lap}^\dagger \beta} = tC\sqrt{n\lambda_1(\mathbf{K}_{Lap}^\dagger)}$. Now, taking expectation over $\sigma$

$$R(\mathcal{H}_{Lap}^\dagger, p) = tC\sqrt{\frac{\lambda_1(\mathbf{K})}{n}} \mathbb{E}_\sigma \left[ \left\| \sum_{i=1}^n \sigma_i \mathbf{v}_i \right\| \right] = tC\sqrt{\frac{\lambda_1(\mathbf{K}_{Lap}^\dagger)}{n}} \mathbb{E}_\sigma \left[ \sqrt{\sigma^\top \mathbf{K}_{Lap}^\dagger \sigma} \right] \tag{7}$$

Using Jensen's inequality and independence of $\sigma$ as before, we can bound the expectation term by $\left(2p \sum_{i=1}^n \mathbf{K}_{Lap_{ii}}^\dagger \right)^{1/2}$. Note that the minimum eigen value of the adjacency $\mathbf{A}$ of a complete graph $K_n$ is $-1$. Thus, the minimum eigen value of $\mathbf{K}_{Lap} = \mathbf{I} - \mathbf{A}/(n-1)$ is $1 + \frac{1}{n-1}$. Thus, $\lambda_1(\mathbf{K}_{Lap}^\dagger) = 1 - \frac{1}{n} \leq 1$. Plugging back in (7) gives

$$R(\mathcal{H}_{Lap}^\dagger, p) = tC\sqrt{\frac{2p}{n} \sum_{i=1}^n \mathbf{K}_{Lap_{ii}}^\dagger} \leq tC\sqrt{2p\lambda_1(\mathbf{K}_{Lap}^\dagger)} \leq tC\sqrt{2p}$$

Thus, for $C, t, p = O(1)$ as in Corollary 4.3 proves the claim. $\qquad \square$

## Section 5

We prove the graph dependent transductive generalization error bound in Theorem 5.1, using concentration inequalities discusses in Section 1.3.

**Proof of Theorem 5.1.** Let $\pi = [\pi_1, \ldots, \pi_n]$ denote a permutation on $[n]$. For any permutation $\pi$, let the first $nf$ nodes be labeled. Let $\ell_i = \ell(y_{\pi_i}, \langle h, \mathbf{u}_{\pi_i} \rangle),\ i \in [n]$; we drop the arguments $h$ and $\pi$, when clear from context. Let $er_S^\ell(h, \mathbf{y}, \pi) = \frac{1}{nf} \sum_{i=1}^{nf} \ell_i$ and $er_{\bar{S}}^\ell(h, \mathbf{y}, \pi) = \frac{1}{n(1-f)} \sum_{i=nf+1}^n \ell_i$. Let $\ell(h, \mathbf{y}, \pi) = [\ell_1, \ldots, \ell_n]$. Let $\bar{\pi} = [1, 2, 3, \ldots, n]$ denote the trivial permutation on [n]. We prove the result in three main steps:

*Step 1:* We introduce a ghost permutation

$$er_{\bar{S}}^\ell(h, \mathbf{y}, \bar{\pi}) = er_S^\ell(h, \mathbf{y}, \bar{\pi}) + er_{\bar{S}}^\ell(h, \mathbf{y}, \bar{\pi}) - er_S^\ell(h, \mathbf{y}, \bar{\pi})$$

$$\leq er_S^\ell(h, \mathbf{y}, \bar{\pi}) + \sup_{h' \in \tilde{\mathcal{H}}_\mathbf{U}} \left[ er_{\bar{S}}^\ell(h', \mathbf{y}, \bar{\pi}) - er_S^\ell(h', \mathbf{y}, \bar{\pi}) \right]$$

$$\leq er_S^\ell(h, \mathbf{y}, \bar{\pi}) + \sup_{h' \in \tilde{\mathcal{H}}_\mathbf{U}} \left[ er_{\bar{S}}^\ell(h', \mathbf{y}, \bar{\pi}) - \mathbb{E}_\pi \left[ er_{\bar{S}}^\ell(h', \mathbf{y}, \pi) \right] \right]$$

$$+ \mathbb{E}_\pi \left[ er_S^\ell(h', \mathbf{y}, \pi) \right] - er_S^\ell(h', \mathbf{y}, \bar{\pi}) \right]$$

The above holds, since $\mathbb{E}_\pi \left[ er_{\bar{S}}^\ell(h', y, \pi) \right] = \mathbb{E}_\pi \left[ er_S^\ell(h', y, \pi) \right]$. Applying Jensen's Inequality,

$$\leq er_S^\ell(h, y, \bar{\pi}) + \Phi(h, y, \bar{\pi}) \tag{8}$$

where $\Phi(h, \mathbf{y}, \bar{\pi}) = \mathbb{E}_\pi \left\{ \sup_{h' \in \tilde{\mathcal{H}}_\mathbf{U}} \left[ er_{\bar{S}}^\ell(h, \mathbf{y}, \bar{\pi}) - er_{\bar{S}}^\ell(h', \mathbf{y}, \pi) + er_S^\ell(h', y, \pi) - er_S^\ell(h, \mathbf{y}, \bar{\pi}) \right] \right\}$.
We invoke the following lemma to bound the supremum

**Lemma 2.1.** For any $\delta > 0$, with probability $\geq 1 - \delta$ over the random permutation $\bar{\pi}$,

$$\Phi(h, \mathbf{y}, \bar{\pi}) \leq \mathbb{E}_{\pi'} \left[ \Phi(h, \mathbf{y}, \pi') \right] + B\sqrt{\frac{2}{nf(1-f)} \log \frac{1}{\delta}}$$

We obtain martingale from the function $\Phi(h, \mathbf{y}, \pi)$ using *Doob's martingale process*. Let $\Pi = [\Pi_1, \ldots, \Pi_n]$ denote a random permutation vector over $[n]$. Let $\Pi_1^i := [\Pi_1, \ldots, \Pi_i]$ and

$$\Phi(h, \mathbf{y}, \Pi_1^i) = \mathbb{E}_{\pi'} \left\{ \sup_{h' \in \tilde{\mathcal{H}}_\mathbf{U}} \left[ er_{\bar{S}}^\ell(h, \mathbf{y}, \Pi_1^i) - er_S^\ell(h', \mathbf{y}, \pi') + er_{\bar{S}}^\ell(h', \mathbf{y}, \pi') - er_S^\ell(h, \mathbf{y}, \Pi_1^i) \right] \right\}$$

where $er_{\bar{S}}^\ell(h, \mathbf{y}, \Pi_1^i) = \frac{\mathbb{1}(i > nf)}{n(1-f)} \sum_{j=nf+1}^{i} \ell_j$ and $er_S^\ell(h, \mathbf{y}, \Pi_1^i) = \frac{1}{nf} \sum_{j=1}^{\min(nf,i)} \ell_j$. Let $\Pi(i, j)$ be the

permutation vector obtained by exchanging the values of $\Pi_i$ and $\Pi_j$. Let $T_0 := \mathbb{E}_{\Pi_1^n} \left[ \ell(h, y, \Pi_1^n) \right]$ and $T_i := \mathbb{E}_{\Pi_1^n} \left[ \ell(h, y, \Pi_1^n) | \Pi_1^i \right], \forall \in [n]$. Clearly $T_0^n$ is a martingale sequence *w.r.t.* $\Pi$. Note that by definition, $T_n = \Phi(h, y, \Pi_1^n)$. Thus, bounding $\tilde{r}$ and using Lemma 1.3 proves the result. Let $\pi_1^n = [\pi_1, \ldots, \pi_n]$ be a specific permutation. For $i \leq nf$

$$r_i(\pi_1^{i-1}) = \sup_{\pi_i} \left\{ T_i : \Pi_1^{i-1} = \pi_1^{i-1}, \Pi_i = \pi_i \right\} - \inf_{\pi_i} \left\{ T_i : \Pi_1^{i-1} = \pi_1^{i-1}, \Pi_i = \pi_i \right\}$$

$$= \sup_{\pi_i, \pi_i'} \left\{ \mathbb{E}_{\Pi_1^n} \left[ \ell(h, \mathbf{y}, \Pi_1^n) \mid \Pi_1^{i-1} = \pi_1^{i-1}, \Pi_i = \pi_i \right] \right.$$

$$\left. - \mathbb{E}_{\Pi_1^n} \left[ \ell(h, \mathbf{y}, \Pi_1^n) \mid \Pi_1^{i-1} = \pi_1^{i-1}, \Pi_i = \pi_i' \right] \right\}$$

$$= \sup_{\pi_i, \pi_i'} \left\{ \mathbb{E}_{j \sim [i+1, n]} \mathbb{E}_{\Pi_1^n} \left[ \ell(h, \mathbf{y}, \Pi_1^n) \mid \Pi_1^{i-1} = \pi_1^{i-1}, \Pi_i = \pi_i, \Pi_j = \pi_i' \right] \right.$$

$$\left. - \mathbb{E}_{j \sim [i+1, n]} \mathbb{E}_{\Pi_1^n} \left[ \ell(h, \mathbf{y}, \Pi_1^n(i, j)) \mid \Pi_1^{i-1} = \pi_1^{i-1}, \Pi_i = \pi_i, \Pi_j = \pi_i' \right] \right\}$$

$$= \sup_{\pi_i, \pi_i'} \left\{ \mathbb{E}_{j \sim [i+1, n]} \mathbb{E}_{\Pi_1^n} \left[ \ell(h, \mathbf{y}, \Pi_1^n) - \ell(h, \mathbf{y}, \Pi_1^n(i, j)) \mid \Pi_1^{i-1} = \pi_1^{i-1}, \Pi_i = \pi_i, \Pi_j = \pi_i' \right] \right\}$$

$$= \sup_{\pi_i, \pi_i'} \left\{ \mathbf{Pr}_{j \sim [i+1, n]} \left\{ j \in [1, nf] \right\} \mathbb{E}_{\Pi_1^n, j \sim [i+1, nf]} \left[ \ell(h, \mathbf{y}, \Pi_1^n) \right. \right.$$

$$\left. - \ell(h, \mathbf{y}, \Pi_1^n(i, j)) \mid \Pi_1^{i-1} = \pi_1^{i-1}, \Pi_i = \pi_i, \Pi_j = \pi_i' \right]$$

$$+ \mathbf{Pr}_{j \sim [i+1, n]} \left\{ j \in [nf + 1, n] \right\} \mathbb{E}_{\Pi_1^n, j \sim [nf+1, n]} \left[ \ell(h, \mathbf{y}, \Pi_1^n) \right.$$

$$\left. \left. - \ell(h, \mathbf{y}, \Pi_1^n(i, j)) \mid \Pi_1^{i-1} = \pi_1^{i-1}, \Pi_i = \pi_i, \Pi_j = \pi_i' \right] \right\}$$

Since $i \leq nf$, $\ell(h, \mathbf{y}, \Pi_1^n) = \ell(h, \mathbf{y}, \Pi_1^n(i, j))$ for $j \in [1, nf]$, thus the first term is zero. Therefore,

$$r_i(\pi_1^{i-1}) \leq \mathbf{Pr}_{j \sim [i+1, n]} (j \in [nf + 1, n]) \frac{B}{nf(1-f)} = \frac{B}{f(n-i)}$$

Also, note that $r_i(\pi_1^{i-1}) = 0$, for $i > nf$. Thus,

$$\tilde{r} = \sup_{\pi_1^n} \sum_{i=1}^{n} (r_i(\pi_1^{i-1}))^2 \leq \sum_{i=1}^{nf} \left( \frac{B}{f(n-i)} \right)^2$$

$$= \left( \frac{B}{f} \right)^2 \sum_{j=n(1-f)}^{n-1} \frac{1}{j^2} \leq \left( \frac{B}{f} \right)^2 \int_{n(1-f)-1/2}^{n-1/2} \frac{1}{j^2} dj$$

$$= \left( \frac{B}{f} \right)^2 \frac{nf}{(n(1-f)-1/2)(n-1/2)} \leq \frac{4B^2}{nf(1-f)}$$

For $1 - f > 0$ and large $n$, $\frac{1}{n-1/2} \leq \frac{2}{n}$ and $\frac{1}{n(1-f)-1/2} \leq \frac{2}{n(1-f)}$. Plugging this in Lemma 1.3 and setting $\epsilon = B\sqrt{\frac{2}{nf(1-f)} \log \frac{1}{\delta}}$ completes the proof of Lemma 2.1.

*Step 2:* Now we concentrate on bounding $\mathbb{E}_{\pi'} [\Phi(h, \mathbf{y}, \pi')]$. For $\mathcal{L}^{\ell}(\tilde{\mathcal{H}}_{\mathbf{U}}) = \{\ell(h, \mathbf{y}, \bar{\pi}) \mid h \in \tilde{\mathcal{H}}_{\mathbf{U}}\}$, we introduce Rademacher variables to prove

**Lemma 2.2.**

$$\mathbb{E}_{\pi'} [\Phi(h, \mathbf{y}, \pi')] \leq R_T(\mathcal{L}^{\ell}(\tilde{\mathcal{H}}_{\mathbf{U}}), f) + O\left( \frac{B}{(1-f)\sqrt{nf}} \right)$$

where $R_T(\mathcal{L}^{\ell}(\tilde{\mathcal{H}}_{\mathbf{U}}), f) = \frac{1}{nf(1-f)} \mathbb{E}_{\sigma} \left[ \sup_{h \in \tilde{\mathcal{H}}_{\mathbf{U}}} \sum_{i=1}^{n} \ell_i \sigma_i \right]$, with $\sigma$ as in 2 with $p = f(1-f)$.

Following notations as in Definition 3, let $P = \sum_{i \in [n]} \mathbb{1}\left[ \tau_{i,1} = -\frac{1}{f} \right]$ and $Q = \sum_{i \in [n]} \mathbb{1}\left[ \tau_{i,2} = \frac{1}{f} \right]$. We prove Lemma 2.2 in four steps:

*Step a: Relating to Pairwise Rademacher variables*

$$R_T(\mathcal{L}^{\ell}(\tilde{\mathcal{H}}_{\mathbf{U}}), f) = \frac{1}{n} \mathbb{E}_{\tau} \left[ \sup_{h \in \tilde{\mathcal{H}}_{\mathbf{U}}} g^{\ell}(\tau, \mathbf{U}, h, \mathbf{y}) \right]$$

where $g^{\ell}(\tau, \mathbf{U}, h, \mathbf{y}) = \sum_{i=1}^{n} (\tau_{i,1} + \tau_{i,2}) \ell_i$. Proof follows from the definition of $\tau$ (Definition 3).

*Step b: Split the Expectation*

$$\mathbb{E}_{\tau} \left[ \sup_{h \in \tilde{\mathcal{H}}_{\mathbf{U}}} g^{\ell}(\tau, \mathbf{U}, h, \mathbf{y}) \right] = \mathbb{E}_{P,Q} \mathbb{E}_{\tau|P,Q} \left[ \sup_{h \in \tilde{\mathcal{H}}_{\mathbf{U}}} g^{\ell}(\tau, \mathbf{U}, h, \mathbf{y}) \right]$$

*Step c: Relating to loss function*

$$\mathbb{E}_{\pi'} [\Phi(h, \mathbf{y}, \pi')] = \mathbb{E}_{\tau|P=\mathbb{E}[P], Q=\mathbb{E}[Q]} \left[ \sup_{h \in \tilde{\mathcal{H}}_{\mathbf{U}}} g^{\ell}(\tau, \mathbf{U}, h, \mathbf{y}) \right]$$

For $\Phi'(p', q', \mathbf{U}, h, h', \mathbf{y}, \pi, \pi') = \frac{1}{1-f} \sum_{i=p'+1}^{n} \ell(y_{\pi_i}, \langle h, \mathbf{u}_{\pi_i} \rangle) - \frac{1}{1-f} \sum_{i=q'+1}^{n} \ell(y_{\pi'_i}, \langle h', \mathbf{u}_{\pi'_i} \rangle) + \frac{1}{f} \sum_{i=1}^{q'} \ell(y_{\pi'_i}, \langle h', \mathbf{u}_{\pi'_i} \rangle) - \frac{1}{f} \sum_{i=1}^{p'} \ell(y_{\pi_i}, \langle h, \mathbf{u}_{\pi_i} \rangle)$, we will prove that for any $p', q' \in [n]$,

$$\mathbb{E}_{\pi,\pi'} [\Phi'(p', q', \mathbf{U}, h, h', \mathbf{y}, \pi, \pi')] = \mathbb{E}_{\tau|P=q', Q=q'} \left[ \sup_{h \in \tilde{\mathcal{H}}_{\mathbf{U}}} g^{\ell}(\tau, \mathbf{U}, h, \mathbf{y}) \right]$$

thus proving the claim for $p' = \mathbb{E}[P]$ and $q' = \mathbb{E}[Q]$ as a special case.

Let $\iota := (\iota_1, \ldots, \iota_n)$, where $\iota_i := (\iota_{i,1}, \iota_{i,2})$ be a random variable taking values of the coefficients of $\ell(y_i, \langle h, u_i \rangle)$ and $\ell(y_i, \langle h', u_i \rangle)$. Let $P' = \sum_{i \in [n]} \mathbb{1}\left[\iota_{i,1} = -\frac{1}{f}\right]$ and $Q' = \sum_{i \in [n]} \mathbb{1}\left[\iota_{i,2} = \frac{1}{f}\right]$. Thus, we can write

$$\mathbb{E}_{\pi,\pi'}\left[\Phi'(p', q', \mathbf{U}, h, h', \mathbf{y}, \pi, \pi')\right] = \mathbb{E}_{\iota | P' = p', Q' = Q'}\left[\sup_{h \in \tilde{\mathcal{H}}_{\mathbf{U}}} g^\ell(\iota, \mathbf{U}, h, \mathbf{y})\right]$$

Note that the distributions $\iota | P' = p', Q' = q'$ and $\tau | P = p', Q = q'$ are identical, thus proving the claim.

*Step d:*

$$\mathbb{E}_{\tau | \mathbb{E}[P], \mathbb{E}[Q]}\left[\sup_{h \in \tilde{\mathcal{H}}_{\mathbf{U}}} g^\ell(\tau, \mathbf{U}, h, y)\right] - \mathbb{E}_{P,Q}\,\mathbb{E}_{\tau | P,Q}\left[\sup_{h \in \tilde{\mathcal{H}}_{\mathbf{U}}} g^\ell(\tau, \mathbf{U}, h, y)\right] = O\left(\frac{B}{1-f}\sqrt{\frac{n}{f}}\right)$$

We use the result from *Step c* to derive the result. Note that for a fixed $q$

$$\mathbb{E}_{\pi,\pi'}\left[\Phi'(p_1, q, \mathbf{U}, h, h', \mathbf{y}, \pi, \pi')\right] - \mathbb{E}_{\pi,\pi'}\left[\Phi'(p_2, q, \mathbf{U}, h, h', \mathbf{y}, \pi, \pi')\right] \le \frac{B}{f(1-f)}|p_1 - p_2|$$

Similar argument holds for $q_1$ and $q_2$, for a fixed $p$. Now, for any $\epsilon \ge 0$,

$$\mathbf{Pr}_{P,Q}\left\{\left|\mathbb{E}_{\tau | P,Q}\left[\sup_{h \in \tilde{\mathcal{H}}_{\mathbf{U}}} g^\ell(\tau, \mathbf{U}, h, \mathbf{y})\right] - \mathbb{E}_{\tau | \mathbb{E}[P], \mathbb{E}[Q]}\left[\sup_{h \in \tilde{\mathcal{H}}_{\mathbf{U}}} g^\ell(\tau, \mathbf{U}, h, \mathbf{y})\right]\right| \ge \epsilon\right\}$$

$$= \mathbf{Pr}_{P,Q}\left\{\left|\mathbb{E}_{\pi,\pi'}\left[\Phi'(P, Q, \mathbf{U}, h, h', \mathbf{y}, \pi, \pi')\right] - \mathbb{E}_{\pi,\pi'}\left[\Phi'(\mathbb{E}[P], \mathbb{E}[Q], \mathbf{U}, h, h', \mathbf{y}, \pi, \pi')\right]\right| \ge \epsilon\right\}$$

$$\le \mathbf{Pr}_{P,Q}\left\{\left|\mathbb{E}_{\pi,\pi'}\left[\Phi'(P, Q, \mathbf{U}, h, h', \mathbf{y}, \pi, \pi')\right] - \mathbb{E}_{\pi,\pi'}\left[\Phi'(P, \mathbb{E}[Q], \mathbf{U}, h, h', \mathbf{y}, \pi, \pi')\right]\right|\right.$$

$$\left. + \left|\mathbb{E}_{\pi,\pi'}\left[\Phi'(P, \mathbb{E}[Q], \mathbf{U}, h, h', \mathbf{y}, \pi, \pi')\right] - \mathbb{E}_{\pi,\pi'}\left[\Phi'(\mathbb{E}[P], \mathbb{E}[Q], \mathbf{U}, h, h', \mathbf{y}, \pi, \pi')\right]\right| \ge \epsilon\right\}$$

$$\le \mathbf{Pr}_{P,Q}\left\{\left|\mathbb{E}_{\pi,\pi'}\left[\Phi'(P, Q, \mathbf{U}, h, h', \mathbf{y}, \pi, \pi')\right] - \mathbb{E}_{\pi,\pi'}\left[\Phi'(P, \mathbb{E}[Q], \mathbf{U}, h, h', \mathbf{y}, \pi, \pi')\right]\right| \ge \epsilon/2\right\}$$

$$+ \mathbf{Pr}_{P,Q}\left\{\left|\mathbb{E}_{\pi,\pi'}\left[\Phi'(P, \mathbb{E}[Q], \mathbf{U}, h, h', \mathbf{y}, \pi, \pi')\right] - \mathbb{E}_{\pi,\pi'}\left[\Phi'(\mathbb{E}[P], \mathbb{E}[Q], \mathbf{U}, h, h', \mathbf{y}, \pi, \pi')\right]\right| \ge \epsilon/2\right\}$$

$$= \mathbf{Pr}_P\left\{|P - \mathbb{E}[P]| \ge \frac{f(1-f)\epsilon}{2B}\right\} + \mathbf{Pr}_Q\left\{|Q - \mathbb{E}[Q]| \ge \frac{f(1-f)\epsilon}{2B}\right\} \le 4\exp\left(-\frac{3f(1-f)^2\epsilon^2}{32B^2 n}\right)$$

The last inequality follows from Bernstein's concentration inequality

$$\mathbf{X} \sim Bin(n, p) \implies \mathbf{Pr}\{|\mathbf{X} - \mathbb{E}[\mathbf{X}]| \ge \epsilon\} \le 2\exp(-3\epsilon^2/8np)$$

and noting that $P, Q$ are binomial random variables $\sim Bin(n, f)$. Note that for any non-negative random variable $\mathbf{Y}$

$$\mathbf{Pr}\{\mathbf{Y} > \epsilon\} \le c_0 \exp(-c_1 \epsilon^2) \implies \mathbb{E}[\mathbf{Y}] \le \sqrt{\ln(c_0 e)/c_1}$$

Applying this for our setting, for $c_0 = 4$, $c_1 = \frac{3f(1-f)^2}{32B^2 n}$ gives

$$\left|\mathbb{E}_{\tau | \mathbb{E}[P], \mathbb{E}[Q]}\left[\sup_{h \in \tilde{\mathcal{H}}_{\mathbf{U}}} g^\ell(\tau, \mathbf{U}, h, \mathbf{y})\right] - \mathbb{E}_{P,Q}\,\mathbb{E}_{\tau | P,Q}\left[\sup_{h \in \tilde{\mathcal{H}}_{\mathbf{U}}} g^\ell(\tau, \mathbf{U}, h, \mathbf{y})\right]\right|$$

$$\le \mathbb{E}_{P,Q}\left|\mathbb{E}_{\tau | \mathbb{E}[P], \mathbb{E}[Q]}\left[\sup_{h \in \tilde{\mathcal{H}}_{\mathbf{U}}} g^\ell(\tau, \mathbf{U}, h, \mathbf{y})\right] - \mathbb{E}_{\tau | P,Q}\left[\sup_{h \in \tilde{\mathcal{H}}_{\mathbf{U}}} g^\ell(\tau, \mathbf{U}, h, \mathbf{y})\right]\right| = O\left(\frac{B}{1-f}\sqrt{\frac{n}{f}}\right)$$

If $f > 1/2$, then by a symmetric argument, we get the bound $O\left(\frac{B}{f}\sqrt{\frac{n}{1-f}}\right)$. Thus we can relax the constraint $f \in (0, 1/2]$; but for simplicity future derivations, we assume $f \leq 1/2$.

Note that combining the above four steps, proves Lemma 2.2.

*Step 3:* From the contraction property of Rademacher complexity, we relate to the function class, and using Theorem 4.1, we prove

**Lemma 2.3.** Given the loss function $\ell$ to be $L$-Lipschitz,

$$R_T(\mathcal{L}^\ell(\tilde{\mathcal{H}}_\mathbf{U}), f) \leq LC\sqrt{\frac{2\lambda_1(\mathbf{K})}{f(1-f)}}$$

We prove $R_T(\mathcal{L}^\ell(\tilde{\mathcal{H}}_\mathbf{U}), f) \leq \frac{L}{f(1-f)}R(\tilde{\mathcal{H}}_\mathbf{U}, f)$ and the result follows from Theorem 4.1, for $t = 1$ and $p = f(1-f)$. We prove the claim by induction. For $h \in \tilde{\mathcal{H}}_\mathbf{U}$, we denote the predictions by $\hat{\mathbf{y}} := \mathbf{U}^\top h$. Thus, $\ell_i = \ell(y_i, \hat{y}_i)$, $\forall i \in [n]$. We prove a more general result, for any $\psi : \tilde{\mathcal{H}}_\mathbf{U} \to \mathbb{R}$, we prove

$$\mathbb{E}_\sigma\left[\sup_{h\in\tilde{\mathcal{H}}_\mathbf{U}}\sum_{i=1}^n \sigma_i\ell_i + \psi(h)\right] \leq \mathbb{E}_\sigma\left[\sup_{h\in\tilde{\mathcal{H}}_\mathbf{U}} L\sum_{i=1}^n \sigma_i\hat{y}_i + \psi(h)\right]$$

The proof is by induction on $k$, such that $0 \leq k \leq n$. For base condition $k = 0$, the proof is trivial. We assume that the inequality holds for $k - 1$ i.e.,

$$\mathbb{E}_{\sigma_1,\ldots,\sigma_{k-1}}\left[\sup_{h\in\tilde{\mathcal{H}}_\mathbf{U}}\sum_{i=1}^{k-1} \sigma_i\ell_i + \psi(h)\right] \leq \mathbb{E}_{\sigma_1,\ldots,\sigma_{k-1}}\left[\sup_{h\in\tilde{\mathcal{H}}_\mathbf{U}}\sum_{i=1}^{k-1} L\sigma_i\hat{y}_i + \psi(h)\right]$$

Now to prove the induction step, consider

$$\mathbb{E}_{\sigma_k}\mathbb{E}_{\sigma_1,\ldots,\sigma_{k-1}}\left[\sup_{h\in\tilde{\mathcal{H}}_\mathbf{U}}\sum_{i=1}^k \sigma_i\ell_i + \psi(h)\right] = p\mathbb{E}_{\sigma_1,\ldots,\sigma_{k-1}}\left[\sup_{h\in\tilde{\mathcal{H}}_\mathbf{U}}\sum_{i=1}^{k-1} \sigma_i\ell_i + \ell_k + \psi(h)\right]$$

$$+ p\mathbb{E}_{\sigma_1,\ldots,\sigma_{k-1}}\left[\sup_{h\in\tilde{\mathcal{H}}_\mathbf{U}}\sum_{i=1}^{k-1} \sigma_i\ell_i - \ell_k + \psi(h)\right] + (1-2p)\mathbb{E}_{\sigma_1,\ldots,\sigma_{k-1}}\left[\sup_{h\in\tilde{\mathcal{H}}_\mathbf{U}}\sum_{i=1}^{k-1} \sigma_i\ell_i + \psi(h)\right]$$

$$\tag{9}$$

where $p = f(1-f)$. We consider bounding the first two terms, since the last term can be easily bounded by the induction assumption. We define functions $\psi'$ and $\psi''$ as follows

$$\mathbb{E}_{\sigma_1,\ldots,\sigma_{k-1}}\left[\sup_{h\in\tilde{\mathcal{H}}_\mathbf{U}}\sum_{i=1}^{k-1} \sigma_i\ell_i + \underbrace{\ell_i + \psi(h)}_{\psi'(h)}\right] + \mathbb{E}_{\sigma_1,\ldots,\sigma_{k-1}}\left[\sup_{h\in\tilde{\mathcal{H}}_\mathbf{U}}\sum_{i=1}^{k-1} \sigma_i\ell_i - \underbrace{\ell_k + \psi(h)}_{\psi''(h)}\right]$$

Now, by induction assumption, we get

$$= \mathbb{E}_{\sigma_1,\ldots,\sigma_{k-1}}\left[\sup_{h\in\tilde{\mathcal{H}}_\mathbf{U}}\left\{L\sum_{i=1}^{k-1} \sigma_i\hat{y}_i + \ell_k + \psi(h)\right\} + \sup_{h\in\tilde{\mathcal{H}}_\mathbf{U}}\left\{L\sum_{i=1}^{k-1} \sigma_i\hat{y}_i - \ell_k + \psi(h)\right\}\right]$$

$$= \mathbb{E}_{\sigma_1,\ldots,\sigma_{k-1}}\left[\sup_{h,h'\in\tilde{\mathcal{H}}_\mathbf{U}}\left\{L\sum_{i=1}^{k-1} \sigma_i(\hat{y}_i + \hat{y}'_i) + \underbrace{\ell(y_k, \hat{y}_k) - \ell(y_k, \hat{y}'_k)}_{} + \psi(h) + \psi(h')\right\}\right]$$

where $\hat{y}' = \mathbf{U}^\top h'$. Using $L$-Lipschitz property of the loss function,

$$\leq \mathbb{E}_{\sigma_1,\ldots,\sigma_{k-1}}\left[\sup_{h,h'\in\tilde{\mathcal{H}}_\mathbf{U}}\left\{\sum_{i=1}^{k-1} L\sigma_i(\hat{y}_i + \hat{y}'_i) + L|\hat{y}_k - \hat{y}'_k| + \psi(h) + \psi(h')\right\}\right]$$

Without loss of generality, we assume $\hat{y}_k \geq \hat{y}'_k$. Suppose $\hat{y}_k \leq \hat{y}'_k$, then we can swap $h$ and $h'$, so that the value of expression increases

$$= \mathbb{E}_{\sigma_1,\ldots,\sigma_{k-1}}\left[\sup_{h,h'\in\tilde{\mathcal{H}}_\mathbf{U}}\left\{\sum_{i=1}^{k-1}L\sigma_i(\hat{y}_i+\hat{y}'_i)+L(\hat{y}_k-\hat{y}'_k)+\psi(h)+\psi(h')\right\}\right]$$

$$= \mathbb{E}_{\sigma_1,\ldots,\sigma_{k-1}}\left[\sup_{h\in\tilde{\mathcal{H}}_\mathbf{U}}\left\{\sum_{i=1}^{k-1}L\sigma_i\hat{y}_i+L\hat{y}_k+\psi(h)\right\}\right]$$

$$+ \mathbb{E}_{\sigma_1,\ldots,\sigma_{k-1}}\left[\sup_{h\in\tilde{\mathcal{H}}_\mathbf{U}}\left\{\sum_{i=1}^{k-1}L\sigma_i\hat{y}_i-L\hat{y}_k+\psi(h)\right\}\right]$$

Plugging this back in (9) and introducing $\sigma_k$ random variable for RHS proves Lemma 2.3.

Combining the results from (8), Lemma 2.1, 2.2 and 2.3 proves the result.

$\square$

Above in an important result connecting empirical error estimate of the unlabeled node set with that of the labelled node set, and an additional class complexity term (Section 4) relating to structural properties of the graph. This result bridges the two domains machine learning and graph theory, which allows us to derive learning theory estimates of empirical unlabeled node set error convergence rate and labeled sample complexity, relating to the famous Lovász $\vartheta$ function of the graph.

## Section 5.1

Using a similar proof technique as in [10], we prove the following result connecting the maximum margin induced by orthonormal representations of graph with the $\vartheta$ function of the graph.

**Proof of Theorem 5.2.** Given a labelled graph $G = (V, E, \mathbf{y})$, $V = [n]$; let $\mathbf{y} \in \mathcal{Y}^n$ be any labelling on the nodes of graph $G$, then note that from Definition 1

$$\vartheta(G) = \min_{\mathbf{U}\in Lab(G)}\min_{\mathbf{c}\in\mathcal{S}^{d-1}}\max_{i\in[n]}\frac{1}{\left(y_i\mathbf{c}^\top\mathbf{u}_i\right)^2}$$

Note an interesting property of orthonormal representations that if $\mathbf{U} \in Lab(G)$, then $\mathbf{U}diag(\varepsilon) \in Lab(G)$ for any $\varepsilon^\top = [\varepsilon_1,\ldots,\varepsilon_n]$ where $\varepsilon_i \in \{+1,-1\}$ $\forall i \in [n]$. Thus, it suffice to consider only those orthonormal representations for which $y_i\mathbf{c}^\top\mathbf{u}_i \geq 0$ $\forall i \in [n]$ holds. For a fixed $\mathbf{c}$, we can rewrite

$$\max_{i\in[n]}\frac{1}{\left(y_i\mathbf{c}^\top\mathbf{u}_i\right)^2} = \left(\min_t t^2 \text{ s.t. } \frac{1}{y_i\mathbf{c}^\top\mathbf{u}_i} \leq t \ \forall i \in [n]\right)$$

Using $\mathbf{w} = t\mathbf{c}$ yields

$$\min_{\mathbf{c}\in\mathcal{S}^{d-1}}\max_{i\in[n]}\frac{1}{\left(y_i\mathbf{c}^\top\mathbf{u}_i\right)^2} = \left(\min_{\mathbf{w}\in\mathbb{R}^d}\|\mathbf{w}\|^2 \text{ s.t. } y_i\mathbf{w}^\top\mathbf{u}_i \geq 1 \ \forall i \in [n]\right)$$

Applying strong duality gives

$$\min_{\mathbf{c}\in\mathcal{S}^{d-1}}\max_{i\in[n]}\frac{1}{\left(y_i\mathbf{c}^\top\mathbf{u}_i\right)^2} = 2\,\omega(\mathbf{K},\mathbf{y})$$

where $\mathbf{K} = \mathbf{U}^\top\mathbf{U} \in \mathcal{K}(G)$. As there is a correspondence between each element of $Lab(G)$ and $\mathcal{K}(G)$ Section 2, minimization of $2\,\omega(\mathbf{K},\mathbf{y})$ over $\mathcal{K}(G)$ is equivalent to computing $\vartheta$ function. $\square$

We illustrate an example of the large class of Labelled SVM-$\vartheta$ graphs, where **LS** labelling approximated the optimal margin within a constant multiplicative factor.

**Claim 3.** Mixture of random graphs defined in Section 5.1 is Labelled SVM-$\vartheta$ graph.

We use the notations as in Section 1.2 We note the following technical lemma the margin induced by **LS** labelling to structural properties of the labelled graph.

**Lemma 2.4.** Let $\mathbf{K}_{LS} = \mathbf{A}/\rho + \mathbf{I}$, $\rho \geq \max\left\{|\lambda_n(\mathbf{A})|, cut(\mathbf{A}, \mathbf{y})\right\}$ be the **LS** labelling of $G$. Let $\mathbf{A}'$ be the adjacency matrix of $G'$ and let $\mathbf{K}'_{LS} = \mathbf{A}'/\rho + \mathbf{I}$. Then,

$$\omega(\mathbf{K}_{LS}, \mathbf{y}) \leq \omega(\mathbf{K}'_{LS}, \mathbf{1}_n) + \frac{1}{2\left(\frac{\rho}{cut(\mathbf{A}, \mathbf{y})} - 1\right)}$$

where $cut(\mathbf{A}, \mathbf{y})$ is as defined in Notations, Section 1.

Such connections are interesting relating geometry and machine learning with graph properties.

*Proof.* First note that $\mathbf{K}'_{LS}$ is a positive semi-definite matrix, since $\mathbf{K}'_{LS}$ is formed by block diagonal sub-matrices of $\mathbf{K}_{LS}$ corresponding to $G_+$ and $G_-$, which are individually positive semi-definite. We analysis the KKT conditions of $\omega(\mathbf{K}_{LS}, \mathbf{1}_n)$ for any **LS** labelling $\mathbf{K}_{LS}$.

$$\alpha_i^* + y_i\left(\sum_{i=1}^{n} K_{ij}\alpha_j^* y_j\right) = 1 + \mu_i^* \tag{10}$$

where $\mu_i^*$ is the lagrange dual variable at the optimal. From convexity, note that there exists $\bar{\alpha} > 0$, such that $\max_{i \in [n]} \alpha_i^* \leq \bar{\alpha}$. From (10), for $\alpha_i^* > 0$

$$\alpha_i^* + \frac{1}{\rho}\sum_{y_i=y_j} A_{ij}\alpha_j^* - \frac{1}{\rho}\sum_{y_i \neq y_j} A_{ij}\alpha_j^* = 1 \implies \alpha_i^* \leq 1 + \frac{1}{\rho}\sum_{y_i \neq y_j} A_{ij}\alpha_j^*$$

Bounding the last inequality using $\bar{\alpha}$ and $cut(\mathbf{A}, \mathbf{y})$ gives $\bar{\alpha} \leq \frac{1}{1 - cut(\mathbf{A}, \mathbf{y})/\rho}$. Thus,

$$\omega(\mathbf{K}_{LS}, \mathbf{y}) = \max_{0 \leq \alpha_i \leq \bar{\alpha}, i \in [n]} \left(\sum_{i=1}^{n} \alpha_i - \frac{1}{2\rho}\sum_{y_i=y_j} \alpha_i\alpha_j A_{ij} + \frac{1}{2\rho}\sum_{y_i \neq y_j} \alpha_i\alpha_j A_{ij}\right)$$

$$\leq \max_{0 \leq \alpha_i \leq 1, i \in [n]} \left(\sum_{i=1}^{n} \alpha_i - \frac{1}{2}\sum_{y_i=y_j} \alpha_i\alpha_j K_{ij}\right) + \frac{cut(\mathbf{A}, \mathbf{y})}{2\rho\left(1 - cut(\mathbf{A}, \mathbf{y})/\rho\right)}$$

Equating the first term by $\omega(\mathbf{K}'_{LS}, \mathbf{1}_n)$ proves the result. $\qquad\square$

**Proof of Claim 3.** From Lemma 1.1, it follows that $\vartheta(G) \leq 2\vartheta\left(G\left(\frac{n}{2}, \frac{1}{2}\right)\right) = \Theta(\sqrt{n})$. The last equality follows from [6]. Also, from Lemma 1.2, $\vartheta(G) \geq \vartheta(G_+) = \Theta(\sqrt{n})$. Thus, $\vartheta(G) = \Theta(G)$.

Given $cut(\mathbf{A}, \mathbf{y}) = c\sqrt{n}$, $c > 1$; note that for $\rho = 2c\sqrt{n}$, $\mathbf{K}_{LS} = \frac{\mathbf{A}}{\rho} + \mathbf{I}$ is a positive semi-definite matrix from the results of [9]. For the notations as in Lemma 1.1, we can show $\omega(\mathbf{K}'_{LS}, \mathbf{1}_n) = \omega(\mathbf{K}_1, \mathbf{1}_{n_1}) + \omega(\mathbf{K}_2, \mathbf{1}_{n_2}) = \Theta(\sqrt{n})$, where $\mathbf{K}_1$ and $\mathbf{K}_2$ are the block diagonal kernel matrices corresponding to graphs $G_+$ and $G_-$ respectively; and the last equality follows from [10]. Thus, using Lemma 2.4 we prove $\omega(\mathbf{K}_{LS}, \mathbf{y}) = \Theta(\sqrt{n})$, and hence the result. $\qquad\square$

The above is a large class of graph family, where **LS** labelling approximated the optimal margin within a constant multiplicative factor

## Section 6

We derive an upper bound on the Lovász $\vartheta$ function of the power law graphs, previously unavailable.
**Claim 4.** For $G(\alpha_1, \alpha_2)$ Power law graphs, where $\left|\{i \in V | d_i = x\}\right| = e^{\alpha_1}/x^{\alpha_2}$ [2]. For $\alpha_2 \in [2.1, 2.5]$, $\vartheta(\bar{G}) = O(\sqrt{n})$

*Proof.* The regime $\alpha_2 \in [2.1, 2.5]$ holds for naturally occurring graphs [1, 4]. Maximum degree of $G(\alpha_1, \alpha_2)$ is given by $d_{max} = e^{\alpha_1/\alpha_2}$ [2]. Note that $\alpha_1 = O(\log n)$ and thus, $d_{max} = O(\sqrt{n})$. Maximum degree is related to chromatic number by $\chi(G) \leq d_{max} + 1$ [15], thus $\chi(G) = O(\sqrt{n})$. Finally, the claim $\vartheta(\bar{G}) = O(\sqrt{n})$ follows from Sandwich Theorem (Section 2). $\qquad\square$

**Claim 5.** Results in Section 6 can be extended to the Inductive setting, both semi-supervised and supervised learning models.

*Proof.* Let $\mathcal{G}$ be any graph family, and let $\mathcal{V}$ be an infinite sequence of labeled node set (Definition 1). Let $\mu$ be the distribution on $\mathcal{V}$. Let $er_{\mathcal{G}}^\ell[h] := \mathbb{E}_{i \sim \mu}[\ell(h(v_i), y_i)]$, the generalization error.

**Semi-supervised setting:** For any $f \in (0, 1)$, let $S$ and $\bar{S}$ be any draw of labelled and unlabelled subgraph from $\mathcal{G}$ respectively. Let $G$ be a graph formed by combining $S$ and $\bar{S}$. Let $\ell$ be any loss function, bounded in $[0, B]$ and $L$-Lipschitz in its second argument. Let $\tilde{\mathcal{H}}_{\mathbf{U}}$, $C$ be as in Theorem 5.1, for any $\mathbf{U} \in Lab(G)$. For any $\delta > 0$, $h \in \tilde{\mathcal{H}}_{\mathbf{U}}$ w.p. $\geq 1 - \delta$, a typical generalization bound is of the form [5]

$$er_{\mathcal{G}}^{\ell}[h] \leq (1-f) \cdot er_{\bar{S}}^{\ell}[h] + f \cdot er_{S}^{\ell}[h] + 2L \cdot \mathbb{E}_{G \sim \mathcal{G}} \big[ R(\tilde{\mathcal{H}}_{\mathbf{U}}, f(1-f)) \big] + O\left( B \sqrt{\frac{1}{n} \log \frac{1}{\delta}} \right)$$

Using Theorem 5.1, we can bound $er_{\bar{S}}^{\ell}[h]$ in-terms of $er_{S}^{\ell}[h]$. The complexity term can be estimated reliably from empirical Rademacher complexity (Theorem 4.1) – using bounded differences inequality [12], w.p. $\geq 1 - \delta/2$

$$\mathbb{E}_{G \sim \mathcal{G}} \big[ R(\mathcal{H}, f(1-f)) \big] \leq R(\tilde{\mathcal{H}}_{\mathbf{U}}, f(1-f)) + tC \sqrt{\frac{\lambda_1(\mathbf{K})}{2} \log \frac{4}{\delta}}$$

Thus, leading to the following generalization bound

$$er_{\mathcal{G}}^{\ell}[h] \leq er_{S}^{\ell}[h] + O\left( LC \sqrt{\frac{\lambda_1(\mathbf{K})}{f} \log \frac{1}{\delta}} + B \sqrt{\frac{1}{nf} \log \frac{1}{\delta}} \right)$$

Using similar proof techniques as in Section 6, we can derive consistency and labeled sample complexity results (similar to Theorem 6.1 and Corollary 6.2) for the semi-supervised setting.

**Supervised setting:** Let $S$ and $\bar{S}$ be any draw of labelled and unlabelled subgraph from $\mathcal{G}$ respectively, for $f = 1/2$. Let $\mathcal{H}$ be any function class. For any $\epsilon > 0$, note the following sandwich theorem on the uniform convergence of generalization error bound

$$\mathbf{Pr}_{S, \bar{S}} \left\{ \sup_{h \in \mathcal{H}} \left| er_{S}^{\ell}[h] - er_{\bar{S}}^{\ell}[h] \right| \geq 2\epsilon \right\} \leq \mathbf{Pr}_{S} \left\{ \sup_{h \in \mathcal{H}} \left| er_{\mathcal{G}}^{\ell}[h] - er_{S}^{\ell}[h] \right| \geq \epsilon \right\}$$

$$\leq 2\mathbf{Pr}_{S, \bar{S}} \left\{ \sup_{h \in \mathcal{H}} \left| er_{S}^{\ell}[h] - er_{\bar{S}}^{\ell}[h] \right| \geq \epsilon/2 \right\}$$

which follows from Symmetrization lemma [14]. Thus, uniform convergence of $er_{\bar{S}}^{\ell}$ to $er_{S}^{\ell}$ in transductive setting is a necessary and sufficient condition for the uniform convergence of the training set error to generalization error, in the supervised setting. Thus, using Theorem 5.1 ($er_{\bar{S}} \to er_{S}$) and Theorem 6.1 ($er_{S} \to 0$), we can extend our analysis to derive consistency and sample complexity results for the inductive setting relating to structural properties of the graph.

$\square$

The above gives us a way to extend the analysis present in the paper to the well known inductive setting. As part of future work, we would also like to analyze the problem of learning on streaming similarity matrices, where similarity information of the data instances is streaming and the task is to accurately predict the binary labels.

## Section 7

We prove generalization bound for the problem of multiple graph transduction

**Proof of Theorem 7.1.** Following similar steps as in Theorem.5.1 we get

$$\frac{1}{|\bar{S}|} \sum_{j \in \bar{S}} \ell^{ramp}(y_j, \hat{y}_j) \leq \frac{1}{|S|} \sum_{i \in S} \ell^{ramp}(y_i, \hat{y}_i) + C \sqrt{\frac{2\lambda_1(\mathbf{K}_{\eta^*})}{f(1-f)}} + \frac{c_1}{1-f} \sqrt{\frac{1}{nf} \log \frac{1}{\delta}}$$

where $c_1 = O(1)$, $\mathbf{K}_{\eta^*} = \sum_{k=1}^m \eta_k^* \mathbf{K}^{(k)}$. Note that $\mathbf{K}_{\eta_*}$ is the orthonormal representation of $G^{\cup}$. Using a result in the proof of Corollary.4.2, $\lambda(\mathbf{K}_{\eta^*}) \leq \vartheta(\bar{G}^{\cup})$ proves the complexity term.

Now, using similar proof technique as in Theorem.6.1,

$$\Psi(\mathbb{K}, \mathbf{y}_S) \leq \min_{k \in [m]} \Lambda_C(\mathbf{K}^{(k)}, \mathbf{y}_S) \leq \min_{k \in [m]} \omega(\mathbf{K}^{(k)}, \mathbf{y}) \left( = \bar{\Psi}(\mathbb{K}, \mathbf{y}) \right)$$

Noting that $C \sum_{i \in S} \ell^{ramp}(y_i, \hat{y}_i) \le \Psi(\mathbb{K}, y_S)$, and $\ell^{ramp}$ is an upper bound on $\ell^{0\text{-}1}$ proves the result.

$\square$

We prove statistical consistency of kernel learning style multiple graph transduction algorithms (6).

**Claim 6.** Similar to Section 6, we can show that if any *one* of the graph families $\mathcal{G}^{(k')}$ of $\mathbb{G}$ obey $\vartheta(G_n^{(k')}) = O(n^c)$, $c \in [0,1)$, $G_n^{(k')} \sim \mathcal{G}^{(k)}$, then the Algorithm optimizing for (6) is $\ell^{0\text{-}1}$-consistent on $\mathbb{G}$ over the class of orthonormal representations.

*Proof.* We will use the notations as in Section 3. Let $\mathbb{K}_n = \{\mathbf{K}_n^{(1)}, \ldots, \mathbf{K}_n^{(m)}\}$, where $\mathbf{K}_n^{(k)} \in \mathcal{K}(G_n^{(k)})$ is the max-margin orthonormal representation associated with $G^{(k)}$. A detailed analysis of the complexity term in Theorem 7.1 gives

$$\lambda_1(\mathbf{K}_{\eta^*}) \le \max_{\eta \in \mathbb{R}_+^m, \|\eta\|_1 = 1} \lambda_1 \Big( \sum_{k=1}^m \eta_k \mathbf{K}_n^{(k)} \Big) \le \max_{\eta \in \mathbb{R}_+^m, \|\eta\|_1 = 1} \sum_{k=1}^m \eta_k \lambda_1 \big( \mathbf{K}_n^{(k)} \big)$$

The last inequality follows from convexity of the spectral norm. At optimality,

$$= \max_{k \in [m]} \lambda_1(\mathbf{K}_n^{(k)}) \le \max_{k \in [m]} \vartheta(\bar{G}_n^{(k)}) = n / \min_{k \in [m]} \vartheta(\bar{G}_n^{(k)})$$

The last two inequalities follows from the results of Corollary 4.2 and the property that for any graph $G - \vartheta(G)\vartheta(\bar{G}) = n$ [11]. Also note that for any $k \in [m]$, $\bar{\Psi}(\mathbb{K}_n, \mathbf{y}) \le \omega(\mathbf{K}_n^{(k)}, \mathbf{y}_n) = \vartheta(G_n^{(k)})/2$, where the last equality follows from Theorem 5.2. Thus, for $k^* = \operatorname{argmin}_{k \in [m]} \vartheta(\bar{G}^{(k)})$, the analysis of multiple graphs $\mathbb{G}$ boils down to the analysis of a single graph family $\mathcal{G}^{(k^*)}$. Plugging the derived results in Theorem 7.1, and setting $\delta = \frac{1}{n}$ and $f = O(1)$ gives for any $C > 0$

$$er_{\bar{S}}^{0\text{-}1}[h_{S_n}] = O \Big( \frac{\vartheta(G_n^{(k^*)})}{Cn} + C \sqrt{\frac{n}{\vartheta(G_n^{(k^*)})}} + \sqrt{\frac{\log n}{n}} \Big) \tag{11}$$

Optimizing for $C$, gives us the error convergence rate $O\Big( \big( \frac{\vartheta(G^{(k^*)})}{n} \big)^{\frac{1}{4}} \Big)$. Thus, bounding $\vartheta(G^{(k^*)}) \le \vartheta(G^{(l)})$, which follows from the optimality of $k^*$ proves the claim. $\square$

**Claim 7.** Given $\mathbb{G}$, such that $\vartheta(G_n) = O(n^c)$, $0 \le c < 1$, $\forall G_n \in \mathcal{G}_c^{(k)}$ for atleast one of the graph families $\mathcal{G}^{(k)}$, $k \in [m]$, there exists $\mathbb{K}$ such that for $C = \Big( \frac{\Psi^2(\mathbb{K}, \mathbf{y})(1-f)}{2^3 n^2 f \vartheta(\bar{G}^{(k^*)})} \Big)^{\frac{1}{4}}$, where $k^* = \operatorname{argmin}_{k \in [m]} \vartheta(\bar{G}^{(k)})$; observing $\frac{1}{2} \Big( \frac{\Psi(\mathbb{K}, \mathbf{y})}{n} \Big)^s$, $0 \le s < 1/3$ fraction of labelled nodes is sufficient for MKL algorithm optimizing for (6) to be $\ell^{0\text{-}1}$-consistent.

Noting that for max-margin orthonormal representations, $\Psi(\mathbb{K}, \mathbf{y}) \le \min_{k \in [m]} \omega(\mathbf{K}^{(k)}, \mathbf{y}) = \vartheta(G^{(k)})/2$ and comparing with Corollary 6.2 shows that combining multiple graphs improves labeled sample complexity than considering individual graphs separately.

*Proof.* From the derivation of Theorem 7.1 and Corollary 6, for $C > 0, f \in (0, 1/2]$; w.p. $\ge 1 - \delta$

$$\frac{1}{|\bar{S}|} \sum_{j \in \bar{S}} \ell^{0\text{-}1}(y_j, \bar{y}_j) \le \frac{1}{C|S|} \Psi(\mathbb{K}, \mathbf{y}) + C \sqrt{\frac{2\vartheta(\bar{G}^{(k^*)})}{f(1-f)}} + \frac{c_1}{1-f} \sqrt{\frac{1}{nf} \log \frac{1}{\delta}}$$

Using $\frac{1}{1-f} \le 2$, setting $\delta = 1/n$ and optimizing over $C$ gives

$$\frac{1}{|\bar{S}|} \sum_{j \in \bar{S}} \ell^{0\text{-}1}(y_j, \bar{y}_j) = O \left( \sqrt{\frac{\Psi(\mathbb{K}, \mathbf{y})}{n} \sqrt{\frac{\vartheta(\bar{G}^{(k^*)})}{f^3}}} + \sqrt{\frac{1}{nf} \log n} \right)$$

Note that $\Psi(\mathbb{K}, \mathbf{y}) \le \omega(\mathbf{K}^{(k^*)}, \mathbf{y}) = \vartheta(G^{k^*})$, where the last inequality follows from Theorem 5.2 for $\mathbb{K}$ being the max-margin orthonormal representations of $\mathbb{G}$. Furthermore, using $\vartheta(G^{(k^*)})\vartheta(\bar{G}^{(k^*)}) = n$ [11] proves the claim. $\square$

Thus, as a consequence of the above result, we can argue that multiple graph transduction improves labeled sample complexity.