[Reviews · NeurIPS 2014]

Submitted by Assigned_Reviewer_7

The authors consider the problem of node classification in graphs, in the transductive setting. For the node classification problem, they argue that orthonormal representation of graphs, i.e. embedding of graphs onto a sphere, where two nodes with no edge between them correspond to two orthonormal vectors on the sphere, is a consistent representation, when used with a certain regularized formulation that is popularly used for vertex labeling. The proof proceeds by considering the function class, learned by the regularized formulation, and using fairly standard techniques to establish generalization bounds.

Bounding the Rademacher complexity of this function class, allows one to provide generalization error bounds. These generalization bounds can be rewritten to involve graph properties, such as the \nu function of graph etc...
This allows the authors to conclude for what kinds of graphs, is the orthonormal representation consistent. One of the neat results that comes up is that for sparse graphs one would need a large fraction of nodes labeled, but for denser graphs even a smaller fraction of nodes being labeled suffices.

The authors then consider the problem of multi-view graph transduction, and provide bounds, by considering the convex combination of kernels corresponding to each of the graphs. They provide generalization error bounds for this multi-view transductive setting.

Simulations are provided which confirm theoretical results.

The paper on the whole is well written, and seems quite novel. There are a few minor typos. e.g. in equation 4, in the definition of $\omega (K,y)$ should the maximization over $\alpha$ be over the set $0\leq \alpha \leq C$? In Alg 1, should $h^{*}$ be replaced with $h_{S_n}$?
It seems from the rebuttal, that these results can also be extended to an inductive setting? Perhaps the authors should add a discussion related to this in the final camera ready version of the paper.
Summary: The paper is well written, and seems to be quite novel. The theoretical results provide an explanation for some intuitively clear results. It would be nice if the authors could provide a small discussion as to how these results (if possible) could be extended to the inductive setting.

Submitted by Assigned_Reviewer_18

The paper presents statistical analyses of graph transduction algorithms, based on orthonormal embedding of graphs introduced by [17]. The main results provided in the paper are:
1) providing a deterministic upper bound on transductive Rademecher complexity in terms of a graph-structural measure (Chromatic number/Lovasz-theta function) for a class of graphs.
2) using that relation to establish graph-dependent generalization bounds for transduction algorithms - the high level proof technique for this part follows [10], but several modification for the current setting are done.
3) establishing consistency of a particular LS labeling algorithm, using the above results for several families of graphs.

A multiple graph extension is also provided, although it is straight-forward given the previous results. The results provided are interesting - particularly the deterministic connection between Rademacher complexity and the structural properties of the graph. The subsequent results are naturally based on this connection. I wonder if there could be tighter upper bounds based on concentration results for \lambda_1(K), that hold with high probability which involves some interpretable parameter of the graph ?

The paper is not well-written which makes following the proofs and interpreting the results time-consuming. Presentation could be improved significantly. I would suggest to move some proofs (or all) in the main paper to the supplementary section and improve the presentation in the main paper (by interpreting the results). Furthermore, there are several places where the style could be made formal (e.g., 163-164; 212-217). Also in algorithm 1, should it be \hat y = U^T h* ? There are several typographical errors that need to be corrected (e.g., supplementary pdf - line 216 - dictribution -> distribution). Also minor changes like providing the original definition of v(G) and then presenting the alternate identity from [14] (lines 124-125) needs to be done.
Summary: The paper presents statistical analyses of graph transduction algorithms, based on orthonormal embedding of graphs.The results provided are interesting, but the presentation needs to be improved significantly.

Submitted by Assigned_Reviewer_36

The authors consider orthonormal representations of graphs. They
present an orthonormal representation called LS labeling which is used
to show consistency. The authors also show K_LS (eq. 6) has a high
Rademacher complexity and claim that it is a better regularizer for
graph transduction. In addition, the authors also explore multiple
graph transduction. Finally, the authors perform a series of
experiments that show the superior performance of their methods.

Overall, this paper contains a significant number of results and is
quite impressive in its scope and detail. The introduction of LS
labeling and associated results are justified both theoretically and
experimentally. In addition, the multiple graph learning is quickly
derived using the methods discussed in the paper. The originality and
significance of the paper is good and the authors clearly outline
their contribution.

A drawback of the paper is the quality and clarity of results. First,
the paper is just too detailed. The authors have packed too much
material in the paper and it feels like the work should be presented
in several papers. Second, the presentation and organization could be
better. The introduction jumps into discussion with notation
explained later. In addition, the paper needs numerous edits to
clarify language. Third, experimental results seem rushed without
much explanation of setup and discussion of results.
Summary: The authors derive and analyze a method for graph-based
transductive learning which performs well and is statistically
consistent. Results are impressive, but presentation and clarity
needs work.
Author Feedback
Author rebuttal: Reviewer 1: (Masked Reviewer ID: Assigned_Reviewer_18)

1.) "A multiple graph extension is also provided, although it is straight-forward given the previous results."
R.) This result helps to develop a sound statistical understanding of why the heuristic of combining graph's inverse Laplacians [3] work well in practice, previously unavailable.
We quote - "If the multiple views are represented by different graphs, the inverse of the graph Laplacian can be regarded as a kernel, and consequently, one may convexly combine graph Laplacians ([3]). The underlying principle of this methodology is unclear however.", as in Dengyong Zhou and Christopher J.C. Burges (2007) on Spectral Clustering and Transductive Learning with Multiple Views, Page 2, Paragraph 1.

2.) "I wonder if there could be tighter upper bounds based on concentration results for \lambda_1(K), that hold with high probability which involves some interpretable parameter of the graph ?"
R.) In the case of G(n,p) random graphs, when p=O(1); we can show that with probability 1-O(1/n), \lambda_1(K_LS)=\Theta(\srqt(n)) (where K_LS is the LS labelling eq.(6)). This gives a constant approximation to the Lovasz theta function of the graph. We will include this result in the final version of the paper.

3.) "I would suggest to move some proofs (or all) in the main paper to the supplementary section and improve the presentation in the main paper. Furthermore, there are several places where the style could be made formal. Typographical errors. Also minor changes like providing the original definition of v(G) and then presenting the alternate identity from [14] (lines 124-125) needs to be done."
R.) We will incorporate the suggested modifications in the main paper.

--------------------------------------------------

Reviewer 2: (Masked Reviewer ID: Assigned_Reviewer_36)

1.) "A drawback of the paper is the quality and clarity of results. First, the paper is just too detailed. The authors have packed too much material in the paper and it feels like the work should be presented in several papers. Second, the presentation and organization could be better. The introduction jumps into discussion with notation explained later. In addition, the paper needs numerous edits to clarify language. Third, experimental results seem rushed without much explanation of setup and discussion of results."
R.) Apologize. We will make appropriate modifications in the final manuscript.

--------------------------------------------------

Reviewer 3: (Masked Reviewer ID: Assigned_Reviewer_7)

1.) "There are a few minor typos"
R.) We will make corrections in the final version of the paper.

2.) "Can these results be extended to an inductive setting?"
R.) Yes. Our results can be extended to inductive (both semi-supervised and supervised) setting as discussed below.

Notations: er_{S} - empirical error on labeled node set. er_{\bar S} - empirical error on unlabeled node set. f - fraction of nodes labeled.

Semi-supervised setting: A typical generalization bound is of the form - expected error \le (1-f)*er_{\bar S} + f*er_{S} + complexity term + slack term.
Using Theorem 5.1, we can bound er_{\bar S} in-terms of er_{S}. For the complexity term, we can use Rademacher complexity (Definition.2, with an additional expectation taken over the observed graph), which can be estimated reliably from empirical Rademacher complexity in Theorem 4.1, using concentration inequalities. Using similar proof techniques as presented in our work, we can derive consistency and labeled sample complexity results (similar to Theorem 6.1 and Corollary 6.2) for the semi-supervised setting, relating to graph-structural measures.

Supervised setting: Our analysis suggests that uniform convergence of er_{\bar S} to er_{S} in transductive setting is a necessary and sufficient condition for the uniform convergence of the empirical labeled set error to the generalization error, in the supervised setting.
Thus, using Theorem 5.1 and Theorem 6.1, we can extend our analysis to derive consistency and graph-dependent sample complexity results.

Thank you for the insightful comment. We will include a short discussion of the above in the final manuscript.